# MEGA: MOVING AVERAGE EQUIPPED GATED ATTENTION

**Xuezhe Ma**[*]
ISI, USC

**Chunting Zhou**[*]
Meta AI

**Xiang Kong**
LTI, CMU

**Junxian He**
SJTU

**Liangke Gui**
LTI, CMU

**Graham Neubig**
LTI, CMU

**Jonathan May**
ISI, USC

**Luke Zettlemoyer**
Meta AI

## ABSTRACT

The design choices in the Transformer attention mechanism, including weak inductive bias and quadratic computational complexity, have limited its application for modeling long sequences. In this paper, we introduce MEGA, a simple, theoretically grounded, single-head gated attention mechanism equipped with (exponential) moving average to incorporate inductive bias of position-aware local dependencies into the position-agnostic attention mechanism. We further propose a variant of MEGA that offers linear time and space complexity yet yields only minimal quality loss, by efficiently splitting the whole sequence into multiple chunks with fixed length. Extensive experiments on a wide range of sequence modeling benchmarks, including the Long Range Arena, neural machine translation, auto-regressive language modeling, and image and speech classification, show that MEGA achieves significant improvements over other sequence models, including variants of Transformers and recent state space models.

## 1 INTRODUCTION

Designing a single unified model to capture long range dependencies in sequential data is a central and challenging problem in sequence modeling. A number of different archtectures have been developed, including convolutional neural networks (CNNs) (Kim, 2014; Strubell et al., 2017), recurrent neural networks (RNNs) (Goller & Kuchler, 1996; Hochreiter & Schmidhuber, 1997), Transformers (Vaswani et al., 2017) and recent state space models (SSMs) (Gu et al., 2022a; Mehta et al., 2022). Among these models, Transformer (Vaswani et al., 2017) has stood out for its impressive empirical success on a wide range of language and vision tasks, including machine translation (Vaswani et al., 2017; Ott et al., 2018), language understanding (Devlin et al., 2019; Liu et al., 2019), image recognition (Dosovitskiy et al., 2020; Touvron et al., 2021) and genetic sequence modeling (Madani et al., 2020; Jumper et al., 2021), mainly because of the conceptually attractive attention mechanism (Bahdanau et al., 2015) which directly models interactions between each pair of input tokens.

However, there are two common drawbacks in this design: i) *weak inductive bias*; and ii) *quadratic computational complexity*. First, the attention mechanism does not assume prior knowledge of the patterns of dependencies between tokens (e.g. positional inductive bias), instead learning to predict the pairwise attention weights directly from data. Second, the cost to compute and store the attention weights is quadratic in the input length. Recent studies have shown the limitations of applying Transformers to longer sequences, both with respect to accuracy and efficiency (Tay et al., 2020).

In this work, we propose a *moving average equipped gated attention mechanism* (MEGA) to solve the two weaknesses simultaneously. The key idea is to incorporate inductive biases into the attention mechanism across the timestep dimension, by leveraging the classic exponential moving average (EMA) approach (Hunter, 1986). EMA captures local dependencies that exponentially decay over time (see Figure 1), and has been widely used in time series data modeling (§2). We introduce a multi-dimensional damped form of EMA with learnable coefficients (§3.1), and subsequently develop the moving average equipped gated attention mechanism by integrating the EMA with a variant of the single-head gated attention (Hua et al., 2022) (§3.2). Theoretically, we show that the single-head gated attention is as expressive as the most commonly used multi-head attention (§3.5). Benefiting

---

[*]Equal Contribution. Contact: `xuezhema@isi.edu`

Table 1: Experimental results of Transformer (XFM), S4 and MEGA on five sequence modeling benchmarks, including long range arena (LRA), machine translation (WMT16 en-de), language modeling (WikiText-103), image classification (ImageNet-1k), raw speech classification (SC-Raw).

|  | **LRA** (Acc. ↑) | **WMT16** (BLEU ↑) | **WT103** (PPL. ↓) | **ImageNet** (Acc. ↑) | **SC** (Acc. ↑) |
|---|---|---|---|---|---|
| XFM | 59.24 | 27.97 | 18.66 | 81.80 | ✗ |
| S4 | 85.86 | – | 20.95 | – | **97.50** |
| MEGA | **88.21** | **29.18** | **18.07** | **82.31** | 96.92 |

from the incorporated moving average mechanism, we further propose a variant of MEGA with linear complexity, named MEGA-chunk, with minimal loss of contextual information (§3.4).

Experimentally, on five sequence modeling tasks across various data types, including long-context sequence modeling, neural machine translation, auto-regressive language modeling, and image and speech classification, MEGA significantly outperforms a variety of strong baseline models, in terms of both effectiveness and efficiency (§4) (see Table 1).

## 2 BACKGROUND

We use $\boldsymbol{X} = \{\mathbf{x}_1, \mathbf{x}_2, \ldots, \mathbf{x}_n\} \in \mathbb{R}^{n \times d}$ to denote a sequence of input representations with length $n$. Let $\boldsymbol{Y} = \{\mathbf{y}_1, \mathbf{y}_2, \ldots, \mathbf{y}_n\} \in \mathbb{R}^{n \times d}$ be the sequence of output representations of each layer with the same length $n$ as the input $\boldsymbol{X}$. In this paper, we assume the representations of the input and output sequences have the same dimension $d$.

### 2.1 SELF-ATTENTION MECHANISM

The traditional self-attention mechanism is a function:

$$\boldsymbol{Y} = \text{Attn}(\boldsymbol{X}) = f\left(\frac{\boldsymbol{Q}\boldsymbol{K}^T}{\tau(\boldsymbol{X})}\right)\boldsymbol{V} \tag{1}$$

where $\text{Attn} : \mathbb{R}^{n \times d} \rightarrow \mathbb{R}^{n \times d}$ is the self-attention function. $\boldsymbol{Q} = \boldsymbol{X}W_q + b_q$, $\boldsymbol{K} = \boldsymbol{X}W_k + b_k$, and $\boldsymbol{V} = \boldsymbol{X}W_v + b_v$ are the sequences of queries, keys and values, with learnable parameters $W_q$, $W_k$, $W_v \in \mathbb{R}^{d \times d}$, and $b_q$, $b_k$, $b_v \in \mathbb{R}^d$. $f(\cdot)$ is an attention function, e.g. the softmax function $f_{\text{softmax}}(\cdot)$ (Bahdanau et al., 2015), or the recently proposed squared ReLU function $f_{\text{relu}^2}(\cdot)$ (So et al., 2021; Hua et al., 2022). $\tau(\boldsymbol{X})$ is a scaling term, which is commonly set to $\tau(\boldsymbol{X}) = \sqrt{d}$ for $f_{\text{softmax}}(\cdot)$, or $\tau(\boldsymbol{X}) = n$ for $f_{\text{relu}^2}(\cdot)$. The commonly used multi-head variant of attention performs the attention function $h$ times in parallel.

We can define a matrix $\boldsymbol{A} = f(\frac{\boldsymbol{Q}\boldsymbol{K}^T}{\tau(\boldsymbol{X})}) \in \mathbb{R}^{n \times n}$ following (1), which is called the *attention matrix*. Since it specifies pairwise dependency weights, the matrix $\boldsymbol{A}$ in principle delivers a flexible and powerful mechanism to learn long-distance dependencies with minimal inductive biases. However, it is in practice a challenging task to recognize all the dependency patterns directly from data, particularly when processing long sequences. Moreover, calculating $\boldsymbol{A}$ with $h$ attention heads takes $O(hn^2)$ time and space, which becomes a significant bottleneck.

### 2.2 EXPONENTIAL MOVING AVERAGE (EMA)

The moving average is a classic approach for sequential data modeling, which has been widely used in time series data to smooth out short-term fluctuations and highlight long-term trends or cycles. The Exponential Moving Average (EMA) (Winters, 1960; Hunter, 1986), a special case of moving average, applies weighting factors that decrease exponentially:

$$\mathbf{y}_t = \boldsymbol{\alpha} \odot \mathbf{x}_t + (1 - \boldsymbol{\alpha}) \odot \mathbf{y}_{t-1}, \tag{2}$$

where $\boldsymbol{\alpha} \in (0, 1)^d$ is the EMA coefficient representing the degree of weighting decrease, and $\odot$ is the element-wise product. A higher $\boldsymbol{\alpha}$ discounts older observations faster (see Figure 1).

Using an EMA places a strong inductive bias on the learning of pairwise dependencies: the dependency weight between two tokens decreases exponentially over time with an input-agnostic decay factor $\boldsymbol{\alpha}$. This property favors local dependencies, and limits long-distance dependencies. Despite the recurrent formulation in (2), the computation of EMA can be represented as $n$ individual convolutions, which can be computed efficiently using fast Fourier transforms (FFTs) (see Appendix A for details).

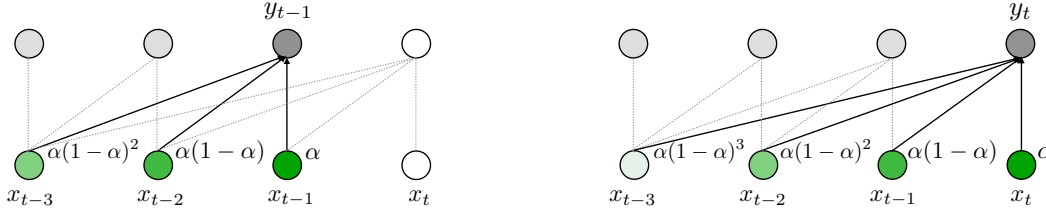

Figure 1: Illustration of the exponential moving average (EMA) approach, which averages the input values $\boldsymbol{X}$ with weights decaying exponentially over timesteps.

## 2.3 WHY COMBINE ATTENTION WITH EMA?

As discussed in Sections 2.1 and 2.2, EMA and attention mechanisms each have their own limitations, despite their wide applications and impressive successes in sequence modeling. By leveraging their properties to complement each other, we propose to embed an EMA into the calculation of the attention matrix $\boldsymbol{A}$. The resulting model enjoys the benefit from strong inductive bias, while maintaining the capacity to learn complex dependency patterns. Moreover, this integration enables the design of a computationally efficient chunk-wise attention mechanism with linear complexity w.r.t sequence length (§3.4).

## 3 MOVING AVERAGE EQUIPPED GATED ATTENTION (MEGA)

In this section, we describe in detail our proposed method, *moving average equipped gated attention* (MEGA). We first introduce *multi-dimensional damped EMA* (§3.1), which is a key component combined with the single-head gated attention in MEGA (§3.2), and discuss the relationship between MEGA and three closely related models: GRU (Cho et al., 2014), Flash (Hua et al., 2022) and S4 (Gu et al., 2022a). Then, we describe the detailed architecture of each MEGA block, including feed-forward and normalization layers (§3.3), and present MEGA-chunk, a variant of MEGA that splits input sequences into fixed chunks, reducing time and space complexity from quadratic to linear (§3.4). At last, we provide theoretical justification for the design of single-head gated attention (§3.5).

### 3.1 MULTI-DIMENSIONAL DAMPED EMA

**Damped EMA.** Previous studies (McKenzie & Gardner Jr, 2010; Svetunkov, 2016) have shown that relaxing the coupled weights of the previous and current observations ($\boldsymbol{\alpha}$ vs. $1 - \boldsymbol{\alpha}$ in (2)) produces robust dependency modeling. Inspired by this, MEGA allows the damping of the influence of the previous time step:

$$\mathbf{y}_t = \boldsymbol{\alpha} \odot \mathbf{x}_t + (1 - \boldsymbol{\alpha} \odot \boldsymbol{\delta}) \odot \mathbf{y}_{t-1}, \tag{3}$$

where $\boldsymbol{\delta} \in (0, 1)^d$ is the damping factor.

**Multi-dimensional Damped EMA.** To further improve the expressiveness of EMA, we introduce a multi-dimensional variant of EMA. Concretely, we first expand each dimension $j \in \{1, 2, \ldots, d\}$ of the input sequence $\boldsymbol{X}$ individually into $h$ dimensions via an expansion matrix $\boldsymbol{\beta} \in \mathbb{R}^{d \times h}$:

$$\mathbf{u}_t^{(j)} = \boldsymbol{\beta}_j \mathbf{x}_{t,j} \tag{4}$$

where $\boldsymbol{\beta}_j \in \mathbb{R}^h$ is the $j$-th row of $\boldsymbol{\beta}$, $\mathbf{u}_t^{(j)} \in \mathbb{R}^h$ is the expanded $h$-dimensional vector for the $j$-th dimension at timestep $t$. Correspondingly, we extend the shape of $\boldsymbol{\alpha}$ and $\boldsymbol{\delta}$ from a one-dimensional vector to a two-dimensional matrix, i.e. $\boldsymbol{\alpha}, \boldsymbol{\delta} \in \mathbb{R}^{d \times h}$, where $\boldsymbol{\alpha}_j, \boldsymbol{\delta}_j \in \mathbb{R}^h$ denote the $j$-th row of $\boldsymbol{\alpha}$ and $\boldsymbol{\delta}$, respectively. Then, for each dimension $j$, the damped EMA is applied to the $h$-dimensional hidden space:

$$\mathbf{h}_t^{(j)} = \boldsymbol{\alpha}_j \odot \mathbf{u}_t^{(j)} + (1 - \boldsymbol{\alpha}_j \odot \boldsymbol{\delta}_j) \odot \mathbf{h}_{t-1}^{(j)}$$
$$\mathbf{y}_{t,j} = \boldsymbol{\eta}_j^T \mathbf{h}_t^{(j)} \tag{5}$$

where $\mathbf{h}_t^{(j)} \in \mathbb{R}^h$ is the EMA hidden state for the $j$-th dimension at timestep $t$. $\boldsymbol{\eta} \in \mathbb{R}^{d \times h}$ is the projection matrix to map the $h$-dimensional hidden state back to 1-dimensional output $\mathbf{y}_{t,j} \in \mathbb{R}$. $\boldsymbol{\eta}_j \in \mathbb{R}^h$ is the $j$-th row of $\boldsymbol{\eta}$. The output $\boldsymbol{Y}$ from (5) is denoted as $\boldsymbol{Y} \triangleq \text{EMA}(\boldsymbol{X})$. Because we do not need to explicitly compute $\mathbf{h}_t^{(j)}$ to get the output $\mathbf{y}_{t,j}$, and the time and space complexity is similar to the standard EMA in (2) (see Appendix A for the details).

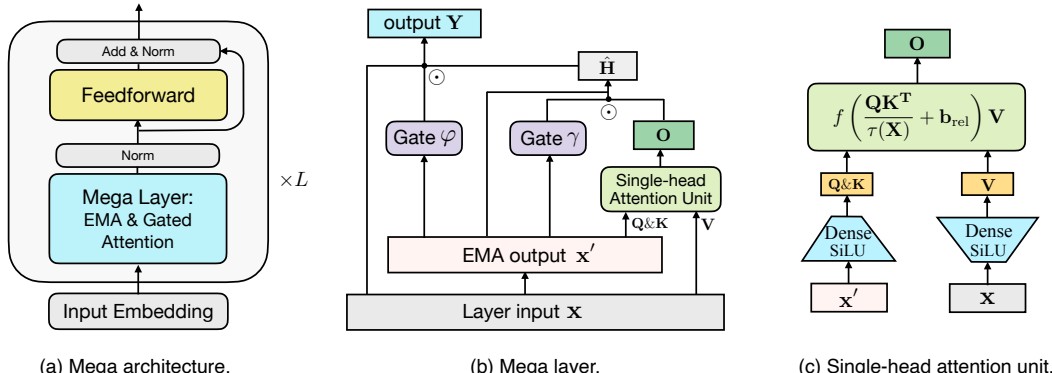

(a) Mega architecture.          (b) Mega layer.          (c) Single-head attention unit.

Figure 2: MEGA – model architecture. (a) shows the overall architecture. (b) illustrates the gated attention sub-layer equipped with EMA, while (c) displays the details of a single-head attention unit.

### 3.2  MOVING AVERAGE EQUIPPED GATED ATTENTION

The gated attention mechanism in MEGA adopts the Gated Recurrent Unit (GRU; Cho et al. (2014)) and Gated Attention Unit (GAU; Hua et al. (2022)) as the backbone architectures, with an EMA-based sub-layer embedded into the calculation of the attention matrix. Formally, we first use the output from (5) to compute the shared representation in GAU:

$$\boldsymbol{X}' = \text{EMA}(\boldsymbol{X}) \qquad\qquad \in \mathbb{R}^{n \times d} \qquad (6)$$

$$\boldsymbol{Z} = \phi_{\text{silu}}(\boldsymbol{X}'W_z + b_z) \qquad\qquad \in \mathbb{R}^{n \times z} \qquad (7)$$

where $\boldsymbol{X}'$ can be regarded as the updated or contextual input, because it encodes contextual information through EMA. $\boldsymbol{Z}$ is the shared representation with $z$ dimensions, with projection matrix $W_z \in \mathbb{R}^{d \times z}$ and bias term $b_z \in \mathbb{R}^z$. $\phi_{\text{silu}}$ is the self-gated activation function (SiLU) (Ramachandran et al., 2017; Elfwing et al., 2018). Following GAU, the query and key sequences are computed by applying per-dimension scalars and offsets to $\boldsymbol{Z}$, and the value sequence is from the original $\boldsymbol{X}$:

$$\boldsymbol{Q} = \boldsymbol{\kappa}_q \odot \boldsymbol{Z} + \boldsymbol{\mu}_q \qquad\qquad \in \mathbb{R}^{n \times z} \qquad (8)$$

$$\boldsymbol{K} = \boldsymbol{\kappa}_k \odot \boldsymbol{Z} + \boldsymbol{\mu}_k \qquad\qquad \in \mathbb{R}^{n \times z} \qquad (9)$$

$$\boldsymbol{V} = \phi_{\text{silu}}(\boldsymbol{X}W_v + b_v) \qquad\qquad \in \mathbb{R}^{n \times v} \qquad (10)$$

where $\boldsymbol{\kappa}_q, \boldsymbol{\mu}_q, \boldsymbol{\kappa}_k, \boldsymbol{\mu}_k \in \mathbb{R}^z$ are the learnable scalars and offsets of queries and keys, respectively. $v$ is the expanded intermediate dimension for the value sequence. The output of attention is:

$$\boldsymbol{O} = f\left(\frac{\boldsymbol{Q}\boldsymbol{K}^T}{\tau(\boldsymbol{X})} + \boldsymbol{b}_{\text{rel}}\right)\boldsymbol{V} \qquad\qquad \in \mathbb{R}^{n \times v} \qquad (11)$$

The graphical specification is displayed in Figure 2 (c). $\boldsymbol{b}_{\text{rel}} \in \mathbb{R}^{n \times n}$ is the relative positional bias. We choose $\boldsymbol{b}_{\text{rel}}$ from existing approaches, including T5 (Raffel et al., 2020), RoPE (Su et al., 2021), TUPE (Ke et al., 2020) and ALiBi (Press et al., 2021).

Subsequently, MEGA introduces the reset gate $\boldsymbol{\gamma}$, the update gate $\boldsymbol{\varphi}$, and computes the candidate activation output $\hat{\boldsymbol{H}}$:

$$\boldsymbol{\gamma} = \phi_{\text{silu}}(\boldsymbol{X}'W_\gamma + b_\gamma) \qquad\qquad \in \mathbb{R}^{n \times v} \qquad (12)$$

$$\boldsymbol{\varphi} = \phi_{\text{sigmoid}}(\boldsymbol{X}'W_\varphi + b_\varphi) \qquad\qquad \in \mathbb{R}^{n \times d} \qquad (13)$$

$$\hat{\boldsymbol{H}} = \phi_{\text{silu}}(\boldsymbol{X}'W_h + (\boldsymbol{\gamma} \odot \boldsymbol{O})U_h + b_h) \qquad\qquad \in \mathbb{R}^{n \times d} \qquad (14)$$

The final output $\boldsymbol{Y}$ is computed with the update gate $\boldsymbol{\varphi}$:

$$\boldsymbol{Y} = \boldsymbol{\varphi} \odot \hat{\boldsymbol{H}} + (1 - \boldsymbol{\varphi}) \odot \boldsymbol{X} \qquad\qquad \in \mathbb{R}^{n \times d} \qquad (15)$$

The graphical architecture of a MEGA sub-layer is visualized in Figure 2 (b).

**Laplace Attention Function.** As mentioned in Section 2.1, the softmax function is the most common choice for the attention function $f(\cdot)$. So et al. (2021) recently introduced the squared ReLU function $f_{\text{relu}^2}(\cdot)$ via architecture search techniques, which has shown faster convergence speed and competitive generalization performance on language tasks (Hua et al., 2022). However, one issue of $f_{\text{relu}^2}(\cdot)$ is that neither its range nor its gradient is bounded, leading to unstable model training (see Appendix C.1 for details). To address this issue, we propose the Laplace attention function:

$$f_{\text{laplace}}(x; \mu, \sigma) = 0.5 \times \left[ 1 + \text{erf}(\frac{x - \mu}{\sigma\sqrt{2}}) \right] \quad (16)$$

where $\text{erf}(\cdot)$ is the error function. $\mu$ and $\sigma$ are two coefficients that we adjust to approximate $f_{\text{relu}^2}$, yielding $\mu = \sqrt{1/2}$ and $\sigma = \sqrt{1/4\pi}$ (The derivations and visualization are provided in Appendix C).

**Relation to and Differences from GRU, Flash and S4.** The computation of the the reset gate $\gamma$, the update gate $\varphi$, and the candidate activation output $\hat{H}$ in (12-14) is reminiscent of GRUs (Cho et al., 2014). The main difference is that in a GRU the two gates are applied between the hidden states of the current and previous timesteps, while in MEGA they are applied between the outputs from EMA and gated attention sub-layers. In addition, the output gating mechanism in (15) is similar to the gated residual connection proposed in Parisotto et al. (2020); Xu et al. (2020).

The computation of the shared representation $Z$, together with the sequences of queries, keys and values in (7-10) are inspired from GAU in Flash (Hua et al., 2022). MEGA integrates EMA into GAU by computing $Z$ in (7) from the EMA output $X'$ rather than the original input $X$, and combining the GAU output with $X'$ for the candidate activation $\hat{H}$ in (14). Experimental gains over Flash demonstrate the effectiveness of this design chice (§4.1).

The multi-dimensional damped EMA can be seen as a simplified variant of a state space model. From this perspective, MEGA is also closely related to S4 (Gu et al., 2022a), a state space model with structured state matrices. S4 leverages the HiPPO framework (Gu et al., 2020) to initialize its low-rank structured state matrices, and the computation of the convolutional kernel in S4 requires complex fast Fourier transformers (FFTs). The EMA sub-layer in MEGA applies diagonalization on the state matrix, which is similar to a concurrent work S4D (Gu et al., 2022b). The differences between the MEGA EMA sub-layer and S4D are: i) S4D parameterizes state matrix in the complex field while EMA uses real numbers; ii) EMA restricts the diagonal elements in the range of $(0, 1)$; iii) EMA does not rely on the HiPPO framework for parameter initialization.

### 3.3 MEGA BLOCKS

The MEGA layer (moving average equipped gated attention) is used as a drop-in-replacement for regular attention in Transformer. It is followed by position-wise feed-forward networks (FFNs) and normalization layers to compose one MEGA block. As the gated residual connection has already been included in (15), we omit the original residual connection and directly apply a normalization layer to $Y$. Concretely,

$$Y = \text{Norm}(\text{Mega}(X))$$
$$Y' = \text{Norm}(\text{FFN}(Y) + Y) \quad (17)$$

where $Y'$ is the output of the MEGA block. The overall architecture of a MEGA block is shown in Figure 2 (a). In Transformer, the hidden dimension of FFNs is usually set to $d_{\text{FFN}} = 4d$. To retain a similar model size with each Transformer block, we reduce the hidden dimension of FFN to $d_{\text{FFN}} = 2d$ and set the expanded dimension $v = 2d$ for the value sequence in (10) throughout this paper, unless specified otherwise.

### 3.4 MEGA-CHUNK: MEGA WITH LINEAR COMPLEXITY

So far we have only focused on introducing stronger inductive bias into attention, which still has quadratic computational complexity. In this section, we propose MEGA-chunk, a variant of MEGA with linear complexity, which simply applies attention to each local chunk of fixed length.

Specifically, we first split the sequences of queries, keys and values in (8-10) into chunks of length $c$. e.g. $Q = \{Q_1, \ldots, Q_k\}$, where $k = n/c$ is the number of chunks.[1] The attention operation

---

[1] Keys and values are split in the same way.

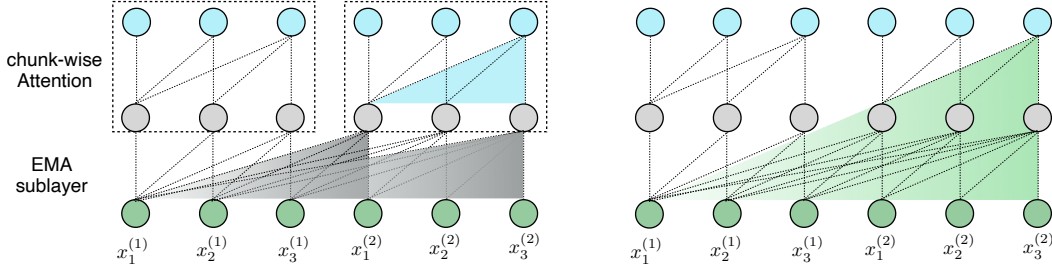

Figure 3: Illustration of the MEGA-chunk model with two chunks of length 3.

in (11) is individually applied to each chunk, yielding linear complexity $O(kc^2) = O(nc)$ w.r.t $n$. However, this method suffers from the critical limitation of losing contextual information from other chunks. Fortunately, the EMA sub-layer in MEGA mitigates this problem by capturing local contextual information near each token, whose outputs are used as the inputs to the attention sub-layer. As a result, the effective context being exploited by chunk-wise attention can go beyond the chunk boundary. Figure 3 illustrates the largest dependency length captured by one MEGA-chunk block.

### 3.5 THEORETICAL JUSTIFICATION OF SINGLE-HEAD GATED ATTENTION

Single-head gated attention has been empirically shown as performant as vanilla multi-head attention Liu et al. (2021); Hua et al. (2022), without any discussions on its theoretical insights. In this section, we provide theoretical justifications of the expressiveness of single-head gated attention. To facilitate subsequent analysis, we simplify the notations of the multi-head attention. Specifically, we denote the sequences of queries, keys and values as the outputs of input transformations:

$$\boldsymbol{Q} = \mathcal{Q}(\boldsymbol{X}), \quad \boldsymbol{K} = \mathcal{K}(\boldsymbol{X}), \quad \boldsymbol{V} = \mathcal{V}(\boldsymbol{X}) \tag{18}$$

where $\mathcal{Q}, \mathcal{K}, \mathcal{V}$ are three transformations, such as linear projections. Let $\boldsymbol{q} \in \boldsymbol{Q} = \{\boldsymbol{q}_1, \dots, \boldsymbol{q}_n\}$ be a single query vector ($\boldsymbol{q} \in \mathbb{R}^d$), and $\boldsymbol{a} = \mathcal{A}(\boldsymbol{q}, \boldsymbol{K})$ denote the corresponding attention weights of $\boldsymbol{q}$, where $\mathcal{A}$ is the attention transformation, i.e. $f(\cdot)$ in (11). For multi-head attention, a common implementation is to split the query into $h$ heads across the model dimension: $\boldsymbol{q} = [\boldsymbol{q}^{(1)}, \dots, \boldsymbol{q}^{(h)}]$, where $\boldsymbol{q}^{(i)} \in \mathbb{R}^{d/h}$, and $i \in \{1, \dots, h\}$ is the query of the $i$-th head. $\boldsymbol{K}$ and $\boldsymbol{V}$ are split in the same way. The attention weight of the $i$-th head is $\boldsymbol{a}^{(i)} = \mathcal{A}(\boldsymbol{q}^{(i)}, \boldsymbol{K}^{(i)})$. Then, the outputs of single-head and multi-head attention are, respectively:

$$\boldsymbol{O}_{\text{SHA}} = \boldsymbol{a}^T \boldsymbol{V} = \begin{bmatrix} \boldsymbol{a}^T \boldsymbol{V}^{(1)} \\ \vdots \\ \boldsymbol{a}^T \boldsymbol{V}^{(h)} \end{bmatrix}, \quad \boldsymbol{O}_{\text{MHA}} = \begin{bmatrix} \boldsymbol{a}^{(1)^T} \boldsymbol{V}^{(1)} \\ \vdots \\ \boldsymbol{a}^{(h)^T} \boldsymbol{V}^{(h)} \end{bmatrix} \tag{19}$$

It is straightforward to see that $\boldsymbol{O}_{\text{MHA}}$ is more expressive than $\boldsymbol{O}_{\text{SHA}}$, because $\boldsymbol{O}_{\text{MHA}}$ leverages $h$ sets of attention weights.

In the single-head gated attention, we introduce a gate vector $\boldsymbol{\gamma} = \mathcal{G}(\boldsymbol{X})$ for each $\boldsymbol{q}$, and the output of single-head gated attention is $\boldsymbol{O}_{\text{SHGA}} = \boldsymbol{O}_{\text{SHA}} \odot \boldsymbol{\gamma}$. The following theorem reveals the equivalence of $\boldsymbol{O}_{\text{SHGA}}$ and $\boldsymbol{O}_{\text{MHA}}$ w.r.t expressiveness (proof in Appendix B):

**Theorem 1** *Suppose the transformation $\mathcal{G}$ is a universal approximator. Then, $\forall \boldsymbol{X}, \exists \boldsymbol{\gamma} = \mathcal{G}(\boldsymbol{X})$ s.t.*

$$\boldsymbol{O}_{\text{SHGA}} = \boldsymbol{O}_{\text{MHA}} \tag{20}$$

Theorem 1 indicates that by simply introducing the gate vector, $\boldsymbol{O}_{\text{SHGA}}$ is as expressive as $\boldsymbol{O}_{\text{MHA}}$. In practice, $\mathcal{G}$ is commonly modeled by a (shallow) neural network, whose universality of approximation has been extensively studied (Hornik et al., 1989; Yarotsky, 2017; Park et al., 2020).

## 4 EXPERIMENTS

To evaluate MEGA, we conduct experiments on five benchmark sequence modeling tasks across various data types. All the numbers with ‡ indicate results from the baseline models replicated by us. More detailed descriptions, results and analysis are provided in Appendix D.

Table 2: (**Long Range Arena**) Accuracy on the full suite of long range arena (LRA) tasks, together with training speed and peak memory consumption comparison on the Text task with input length of 4K. ‡ indicates results replicated by us.

| Models | ListOps | Text | Retrieval | Image | Pathfinder | Path-X | Avg. | Speed | Mem. |
|---|---|---|---|---|---|---|---|---|---|
| XFM | 36.37 | 64.27 | 57.46 | 42.44 | 71.40 | ✗ | 54.39 | – | – |
| XFM‡ | 37.11 | 65.21 | 79.14 | 42.94 | 71.83 | ✗ | 59.24 | 1× | 1× |
| Reformer | 37.27 | 56.10 | 53.40 | 38.07 | 68.50 | ✗ | 50.67 | 0.8× | 0.24× |
| Linformer | 35.70 | 53.94 | 52.27 | 38.56 | 76.34 | ✗ | 51.36 | 5.5× | 0.10× |
| BigBird | 36.05 | 64.02 | 59.29 | 40.83 | 74.87 | ✗ | 55.01 | 1.1× | 0.30× |
| Performer | 18.01 | 65.40 | 53.82 | 42.77 | 77.05 | ✗ | 51.41 | **5.7×** | **0.11×** |
| Luna-256 | 37.98 | 65.78 | 79.56 | 47.86 | 78.55 | ✗ | 61.95 | 4.9× | 0.16× |
| S4-v1 | 58.35 | 76.02 | 87.09 | 87.26 | 86.05 | 88.10 | 80.48 | – | – |
| S4-v2 | 59.60 | 86.82 | 90.90 | 88.65 | 94.20 | 96.35 | 86.09 | – | – |
| S4-v2‡ | 59.10 | 86.53 | 90.94 | 88.48 | 94.01 | 96.07 | 85.86 | 4.8× | 0.14× |
| MEGA | **63.14** | **90.43** | **91.25** | **90.44** | **96.01** | **97.98** | **88.21** | 2.9× | 0.31× |
| MEGA-chunk | 58.76 | 90.19 | 90.97 | 85.80 | 94.41 | 93.81 | 85.66 | 5.5× | 0.13× |

## 4.1 LONG-CONTEXT SEQUENCE MODELING

We begin our experiments with an evaluation on the Long Range Arena (LRA) benchmark recently introduced by Tay et al. (2021), which is designed for evaluating sequence models under the long-context scenario. This benchmark has six tasks: ListOps (Nangia & Bowman, 2018), byte-level text classification (Text; Maas et al. (2011)), byte-level document retrieval (Retrieval; Radev et al. (2013)), image classification on sequences of pixels (Image; Krizhevsky et al. (2009)), Pathfinder (Linsley et al., 2018) and its extreme long version (Path-X; Tay et al. (2021)). These tasks consist of input sequences ranging from 1K to 16K tokens and span across a variety of data types and modalities.

Table 2 compares MEGA against several baselines, including Transformer and its efficient variants, and the state-of-the-art S4 models (both version 1 (Gu et al., 2022a) and version 2 (Gu et al., 2022b)).[2] To ensure fair comparison, we adjust the number of layers and model dimensions on each task so that MEGA has similar number of parameters with S4-v1. For each experiment, we report the average over 5 runs with different random seeds.

On all the six tasks, MEGA substantially outperforms all the baselines. We also evaluate MEGA-chunk on each task, by setting the chunk size $c = 128$ for all the tasks, except Path-X where $c = 4096$. We observe that MEGA-chunk consistently performs well, particularly on the three language tasks. We also examine the speed and memory efficiency of MEGA on the byte-level classification task with the input length of 4K. MEGA-chunk is highly efficient, which is about 5.5 times faster and consumes only 13% as much memory as the vanilla Transformer. In addition, MEGA with full attention field is also much more efficient than Transformer, benefiting from single-head gated attention.

**Analysis of Multi-dimensional Damped EMA.** To demonstrate the effectiveness of the multi-dimensional damped EMA component in MEGA, we performs ablation studies on two LRA tasks — byte-level text classification (Text) and image classification on sequences of pixels (Image). We train MEGA models with EMA dimension $h \in \{0, 1, 2, 4, 8, 16, 32\}$, where $h = 0$ indicates removing the EMA component. From the left figure in Figure 4, we see that without the EMA component, model accuracy on both the two tasks declines rapidly. Meanwhile, the significant improvements of MEGA with a single dimensional EMA ($h = 1$) demonstrate the importance of EMA inductive bias.

**Analysis of Chunk Size.** We further analyze the impact of chunk size $c$ on the same two tasks, by varying $c \in \{16, 32, 64, 128, 256, 512, \infty\}$, where $\infty$ indicates the original MEGA without chunking. The right figure in Figure 4 shows that image data is more sensitive to chunk size than text data. On the Text task, MEGA-chunk with even a small chunk size $c = 16$ is able to achieve around 90% accuracy. On the Image task, MEGA-chunk with $c = 16$ achieves around 75% accuracy, which is still much better than the vanilla Transformer model.

---

[2]The S4-v2 used larger model sizes and better-tuned hyper-parameters than S4-v1. We have also experimented with SRU++ (Lei, 2021) on Pathfinder but failed to converge on this dataset after tuning hyperparameters.

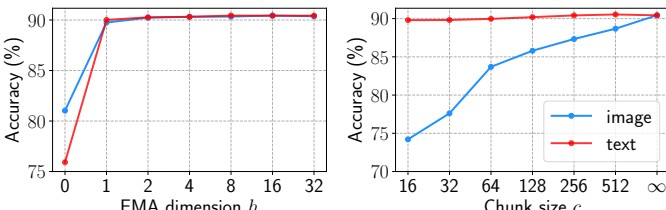

| | Text | Image |
|---|---|---|
| softmax | **90.43** | 89.87 |
| relu[2] | 90.08 | 90.22 |
| laplace | 90.22 | **90.43** |

Figure 4: Ablations on EMA dimension and chunk size.

Table 3: Attention functions.

**Analysis of Attention Functions.** Finally, we evaluate performance with different attention functions. Table 3 shows the accuracy of the three attention functions. On text data softmax obtains the best accuracy, while on image data it performs the worst. The laplace function achieves the best accuracy on Image and also competitive result on Text, being consistently better than relu[2]. In the following experiments we use softmax for language tasks and laplace for vision and speech ones.

## 4.2 RAW SPEECH CLASSIFICATION

To evaluate the capability of MEGA on the long-range modeling of speech signals, we apply MEGA to classify raw speech (with length 16000), rather than using traditional preprocessing (e.g. convert to MFCC features). Following Gu et al. (2022a), we perform speech classification on the SC10 subset of the Speech Commands dataset (Warden, 2018). We experiment with the MEGA-chunk variant with $c = 1000$, since the computation of MEGA and Transformer can not fit in GPU memory. As shown in Table 4, our MEGA-chunk (base) model with 300K parameters is able to achieve an accuracy of 96.92 that is slightly worse than 97.50 from the state-of-the-art method S4,[3] while by adding 0.18M parameters our MEGA-chunk (big) model performs comparably well with S4.

## 4.3 AUTO-REGRESSIVE LANGUAGE MODELING

We evaluate MEGA on two established language modeling benchmarks — WikiText-103 (Merity et al., 2017) and enwik8 (Hutter, 2006). WikiText-103 is a word-level language modeling dataset containing 103M training tokens from Wikipedia articles. Following previous work (Baevski & Auli, 2018), we adopt adaptive softmax and input embeddings and use a vocabulary of 260K tokens. Enwik8 is a character-level language modeling benchmark that has 100M tokens of unprocessed Wikipedia articles and a vocabulary size of about 200. At test time, we split the test data into segments and process each segment sequentially. In Table 5, we compare with previous top-performing models that are designed to take advantage of longer context, including Transformers (Baevski & Auli, 2018; Al-Rfou et al., 2019) (XFM-adaptive), Transformer-XL (Dai et al., 2019) (XFM-XL) and S4 (Gu et al., 2022a). On both WikiText-103 and enwik8, we obtain very competitive results, outperforming baselines by a large margin while enjoying much faster ($9\times$) inference speed compared to the Transformer model. MEGA can also naturally achieve **length extrapolation** at inference time to any sequences that are longer than those seen during training due to the recurrent design of the EMA layer. In addition, we can extrapolate to a longer chunk size for MEGA attention due to the use of rotary positional embeddings for training (Su et al., 2021). The details are provided in in Appendix D.4.

## 4.4 NEURAL MACHINE TRANSLATION

To evaluate MEGA on sequence-to-sequence modeling, we conduct experiments on a standard machine translation benchmark, WMT 2016 English-German news translation (WMT'16), consisting of 4.5M sentence pairs of training data, following Ott et al. (2018). The MEGA models closely follow the architecture of Transformer-base: 6 encoder and decoder layers with model dimension $d = 512$.

Table 7 presents the BLEU scores on the test sets of WMT'16 from two directions: EN→DE and DE→EN. For each experiment, we report the average of both tokenized and SacreBLEU (Post, 2018) scores with 5 different random seeds. MEGA-base significantly outperforms Transformer-base by over 1.1 BLEU. We also report results of MEGA with the Laplace attention function, which slightly but consistently underperforms Softmax.

---

[3]Our S4 number is obtained by running the official S4 code and is a bit worse than the original reported number (98.32), due to different data splits — the order of `os.listdir` is not deterministic across machines.

Table 4: (**SC-Raw**) Accuracy on Speech Commands.

| | SC-Raw | |
| Model | #Param. | Acc. |
|---|---|---|
| XFM | 786K | ✗ |
| S4‡ | 300K | **97.50** |
| MEGA (base) | 300K | 96.92 |
| MEGA (big) | 476K | 97.30 |

Table 5: (**Language Modeling**) PPL (↓) on WikiText-103 and bpc (↓) on enwik8.

| | WikiText-103 | | | enwik8 | |
| Model | #Param. | PPL | Speed | #Param. | bpc |
|---|---|---|---|---|---|
| XFM-adaptive | 247M | 18.66 | 5.6k t/s | - | - |
| XFM-XL | 257M | 18.30 | - | 41M | 1.06 |
| S4 | 249M | 20.95 | - | - | - |
| MEGA | 252M | **18.07** | 48k t/s | 39M | **1.02** |

Table 6: (**ImageNet**) Top-1 accuracy.

| Model | Img. size | #Param. | Acc. |
|---|---|---|---|
| ResNet-152 | $224^2$ | 60M | 78.3 |
| VIT-B | $384^2$ | 86M | 77.9 |
| DeiT-B | $224^2$ | 86M | 81.8 |
| MEGA | $224^2$ | 90M | **82.3** |

Table 7: (**WMT'16**) Test BLEU scores.

| | EN-DE | | DE-EN | |
| Model | Token. | Sacre. | Token. | Sacre. |
|---|---|---|---|---|
| XFM-base | 27.30 | – | – | – |
| XFM-base‡ | 27.97 | 27.33 | 31.92 | 31.33 |
| MEGA-softmax | **29.18** | **28.47** | **32.90** | **32.35** |
| MEGA-laplace | 28.95 | 28.27 | 32.81 | 32.22 |

## 4.5 IMAGE CLASSIFICATION

To evaluate MEGA on a large-scale image classification task, we conduct experiments on the Imagenet-$1k$ (Deng et al., 2009) dataset, which consists of 1.28M training images and 50K validation images from 1000 classes. Top-1 accuracy on the validation set is reported in Table 6 to assess various models. MEGA obtains about $0.5\%$ accuracy improvement over DeiT-B (Touvron et al., 2021). We mostly follow DeiT's approach of applying several data augmentation and regularization methods that facilitate the training process, including Cutmix (Yun et al., 2019), Mixup (Zhang et al., 2017), stochastic depth (Huang et al., 2016), repeated augmentation (Hoffer et al., 2020), Rand-Augment (Cubuk et al., 2020), and random erasing (Zhong et al., 2020). These methods were highly tuned towards optimizing the performance of DeiT, which might be sub-optimal for MEGA. Exploring the optimal data augmentation and regularization methods for MEGA is an interesting direction for future work.

## 5 RELATED WORK

To incorporate stronger inductive bias into the attention mechanism, one research direction focuses on developing advanced positional encoding methods, including absolute and relative positional embeddings (Vaswani et al., 2017; Huang et al., 2020; Ke et al., 2020), and relative positional biases (Su et al., 2021; Press et al., 2021). Another line of research combines the attention mechanism with other neural architectures with intrinsic strong inductive bias, such as convolutional (Gehring et al., 2017; Dai et al., 2021) and recurrence (Dai et al., 2019; Rae et al., 2020; Lei, 2021). Recently, many advanced variants of Transformer models (*'xformers'*) (Tay et al., 2020; 2021) have emerged to improve the time and memory efficiency. Popular techniques include sparse attention patterns (Parmar et al., 2018; Beltagy et al., 2020; Kitaev et al., 2020), low-rank approximations of the attention matrix (Wang et al., 2020; Ma et al., 2021), and approximations through kernelization (Choromanski et al., 2020; Peng et al., 2021). Although these models demonstrate better *asymptotic* complexity for long sequences, their efficiency gains are less prominent for moderate length sequences and their performance lags behind Transformers with regular attention. In addition, as EMA and more general state space models such as S4 can be regarded as a convolution transform with kernel size equal to the sequence length, MEGA is also relevant with CNNs with continuous kernels (Romero et al., 2022).

## 6 CONCLUSION

We have introduced MEGA, a simple, efficient and effective neural architecture used as a drop-in replacement for regular multi-head attention. By leveraging the classic exponential moving average (EMA) approach, MEGA is capable of incorporating stronger inductive biases into the attention mechanism. Moreover, the EMA approach enables the design of MEGA-chunk, an efficient variant of MEGA with linear complexity. On five sequence modeling tasks across various data types, MEGA achieves impressive improvements over a variety of strong baselines. This leads to a potential direction of future work to apply MEGA for multi-modality modeling.

## ACKNOWLEDGEMENT

We thank the anonymous reviewers for their comments. This material is based on research sponsored by Air Force Research Laboratory (AFRL) under agreement number FA8750-19-1-1000. The U.S. Government is authorized to reproduce and distribute reprints for Government purposes notwithstanding any copyright notation therein.

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

APPENDIX: MEGA: MOVING AVERAGE EQUIPPED GATED ATTENTION

## A  EFFICIENT COMPUTATION OF MULTI-DIMENSIONAL DAMPED EMA

Note that the computation of the multi-dimensional damped EMAs of different dimensions are entirely independent of each other. Without loss of generality, we set $d = 1$ and omit the dimension index $j$ in the following formulations. We denote the initial hidden state as $\boldsymbol{h}_0$. The multi-dimensional damped EMA defined in (5) can be vectorized into the following formulation:

$$\mathbf{h}_t = \boldsymbol{\alpha} \odot \mathbf{u}_t + (1 - \boldsymbol{\alpha} \odot \boldsymbol{\delta}) \odot \mathbf{h}_{t-1} \tag{21}$$

$$\mathbf{y}_t = \boldsymbol{\eta}^T \mathbf{h}_t \tag{22}$$

where $\boldsymbol{\alpha}$, $\boldsymbol{\delta}$, and $\boldsymbol{\eta} \in \mathbb{R}^h$. $\mathbf{u}_t = \boldsymbol{\beta} \mathbf{x}_t \in \mathbb{R}^h$ and $\mathbf{h}_t \in \mathbb{R}^h$ is the EMA hidden state at timestep $t$.

Let's denote $\boldsymbol{\phi} = 1 - \boldsymbol{\alpha} \odot \boldsymbol{\delta}$. Then, unrolling the above two equations explicitly yields:

$$\mathbf{h}_1 = \boldsymbol{\phi} \odot \mathbf{h}_0 + \boldsymbol{\alpha} \odot \boldsymbol{\beta} \mathbf{x}_1 \qquad \mathbf{h}_2 = \boldsymbol{\phi}^2 \odot \mathbf{h}_0 + \boldsymbol{\phi} \odot \boldsymbol{\alpha} \odot \boldsymbol{\beta} \mathbf{x}_1 + \boldsymbol{\alpha} \odot \boldsymbol{\beta} \mathbf{x}_2 \qquad \ldots$$

$$\mathbf{y}_1 = \boldsymbol{\eta}^T \boldsymbol{\phi} \odot \mathbf{h}_0 + \boldsymbol{\eta}^T \boldsymbol{\alpha} \odot \boldsymbol{\beta} \mathbf{x}_1 \quad \mathbf{y}_2 = \boldsymbol{\eta}^T \boldsymbol{\phi}^2 \odot \mathbf{h}_0 + \boldsymbol{\eta}^T \boldsymbol{\phi} \odot \boldsymbol{\alpha} \odot \boldsymbol{\beta} \mathbf{x}_1 + \boldsymbol{\eta}^T \boldsymbol{\alpha} \odot \boldsymbol{\beta} \mathbf{x}_2 \quad \ldots$$

This can be written into a vectorized formula:

$$\mathbf{y}_t = \boldsymbol{\eta}^T \boldsymbol{\phi}^t \odot \mathbf{h}_0 + \boldsymbol{\eta}^T \boldsymbol{\phi}^{t-1} \odot \boldsymbol{\alpha} \odot \boldsymbol{\beta} \mathbf{x}_1 + \ldots + \boldsymbol{\eta}^T \boldsymbol{\alpha} \odot \boldsymbol{\beta} \mathbf{x}_t \tag{23}$$

$$\mathbf{y} = \mathcal{K} * \mathbf{x} + \boldsymbol{\eta}^T \boldsymbol{\phi}^t \odot \mathbf{h}_0 \tag{24}$$

where $*$ is the convolution transform with kernel $\mathcal{K} \in \mathbb{R}^n$:

$$\mathcal{K} = \left( \boldsymbol{\eta}^T (\boldsymbol{\alpha} \odot \boldsymbol{\beta}), \ \boldsymbol{\eta}^T (\boldsymbol{\phi} \odot \boldsymbol{\alpha} \odot \boldsymbol{\beta}), \ \ldots, \ \boldsymbol{\eta}^T (\boldsymbol{\phi}^t \odot \boldsymbol{\alpha} \odot \boldsymbol{\beta}) \right) \tag{25}$$

In the proposed multi-dimensional damped EMA, $\mathcal{K}$ can be efficiently computed by the Vandermonde product. With $K$ provided, the output $\mathbf{y}$ in (24) can be computed efficiently with FFTs.

## B  PROOF OF THEOREM 1

**Proof**  We split $\boldsymbol{\gamma}$ into $h$ heads in the same way as $\boldsymbol{Q}$, $\boldsymbol{K}$, and $\boldsymbol{V}$:

$$\boldsymbol{\gamma} = \left[ \begin{array}{c} \boldsymbol{\gamma}^{(1)} \\ \vdots \\ \boldsymbol{\gamma}^{(h)} \end{array} \right]$$

Then we have

$$\boldsymbol{O}_{\text{SHGA}} = \boldsymbol{a}^T \boldsymbol{V} \odot \boldsymbol{\gamma} = \left[ \begin{array}{c} \boldsymbol{a}^T \boldsymbol{V}^{(1)} \odot \boldsymbol{\gamma}^{(1)} \\ \vdots \\ \boldsymbol{a}^T \boldsymbol{V}^{(h)} \odot \boldsymbol{\gamma}^{(h)} \end{array} \right]$$

To prove Theorem 1, we need to find $\boldsymbol{\gamma}$ such that

$$\boldsymbol{a}^T \boldsymbol{V}^{(i)} \odot \boldsymbol{\gamma}^{(i)} = \boldsymbol{a}^{(i)^T} \boldsymbol{V}^{(i)} \iff \boldsymbol{\gamma}^{(i)} = \boldsymbol{a}^{(i)^T} \boldsymbol{V}^{(i)} \oslash \boldsymbol{a}^T \boldsymbol{V}^{(i)}, \ \forall i \in \{1, \ldots, h\},$$

where $\oslash$ is the element-wise divide operation. Since $\mathcal{G}(\boldsymbol{X})$ is a universal approximator and $\boldsymbol{Q}$, $\boldsymbol{K}$, $\boldsymbol{V}$ and $\boldsymbol{a}$ are all transformed from $\boldsymbol{X}$, $\boldsymbol{\gamma}$ can theoretically recover $\boldsymbol{a}^{(i)^T} \boldsymbol{V}^{(i)} \oslash \boldsymbol{a}^T \boldsymbol{V}^{(i)}, \ \forall \boldsymbol{X}$. ∎

## C  LAPLACE ATTENTION FUNCTION

To approximate the squared ReLU function with the Laplace function in (16), we need to select proper coefficients $\mu$ and $\sigma$. We derive the values of $\mu$ and $\sigma$ by solving the following two equations at $x = \sqrt{2}$:

$$f_{\text{relu2}}(\sqrt{2}) = f_{\text{laplace}}(\sqrt{2}) \tag{26}$$

$$f'_{\text{relu2}}(\sqrt{2}) = f'_{\text{laplace}}(\sqrt{2}) \tag{27}$$

The Eq. (26) delivers $\mu = \sqrt{1/2}$ and Eq. 27 subsequently provides $\sigma = \sqrt{1/4\pi}$. Figure 5 visualizes the two functions.

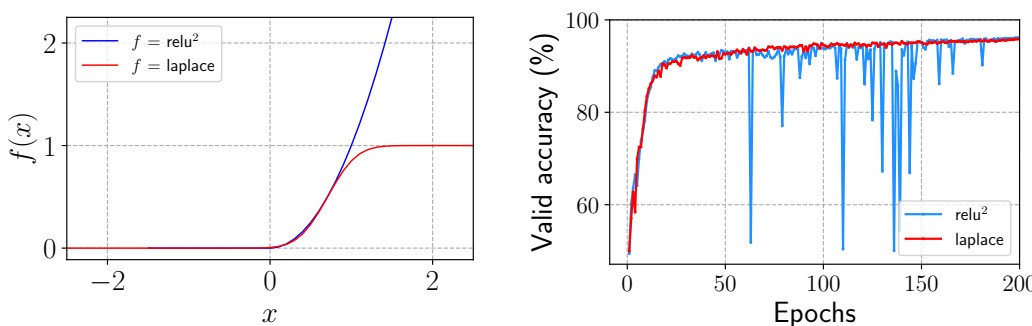

Figure 5: Laplace vs. ReLU$^2$.

## C.1   STABILITY: LAPLACE VS. SQUARED RELU

Besides performance improvements, we also investigate the stability of the two attention functions. We conduct experiments on the LRA Pathfinder task with MEGA models with the two functions. Figure 5 presents the accuracy on the validation set across training epochs. We observe that Laplace is much more stable than ReLU$^2$.

## D EXPERIMENTAL DETAILS

### D.1 DETAILS OF MEGA IMPLEMENTATION

**Uni-directional vs. Bi-directional EMA.** In all the experiments, bi-directional EMA is applied when MEGA is used in the encoder mode while uni-directional one is used in the decoder mode with auto-regressive property.

**Relative Positional Bias.** We only used ROPE (Su et al., 2021) for the language modeling tasks because we conducted experiments of length extrapolation to longer attention chunks at inference time D.4. For other tasks, we simply use very simple learnable relative biases, i.e. one learnable parameter for each relative positional distance (e.g. -3, +4).

### D.2 LONG RANGE ARENA (LRA)

For all tasks, we closely follow Tay et al. (2020) for details such as data preprocessing, data split, etc. The hyper-parameters of MEGA models on these tasks are listed in Table 8.

Table 8: Hyper-parameters of MEGA models on LRA and raw speech classification tasks. BSZ is batch size, LR is learning rate and WD is weight decay. BN, LN and SN refer to Batch Normalization, Layer Normalization and Scale Normalization.

| Task | Depth | $d_{\text{model}}$ | $d_{\text{FFN}}$ | $z$ | $v$ | $h$ | Attn-FN | Norm | Pre-norm | BSZ | LR | Dropout | WD | Epochs |
|------|-------|-------|------|-----|-----|-----|---------|------|----------|-----|-----|---------|-----|--------|
| **ListOps** | 6 | 80 | 160 | 64 | 160 | 16 | softmax | LN | False | 64 | 0.001 | 0.1 | 0.01 | 60 |
| **Text** | 4 | 128 | 256 | 64 | 256 | 16 | softmax | SN | False | 50 | 0.004 | 0.1 | 0.01 | 50 |
| **Retrieval** | 6 | 128 | 256 | 64 | 256 | 16 | softmax | SN | False | 64 | 0.003 | 0.1 | 0.04 | 40 |
| **Image** | 8 | 160 | 320 | 96 | 320 | 16 | laplace | BN | True | 50 | 0.01 | 0.0 | 0.02 | 200 |
| **Pathfinder** | 6 | 128 | 256 | 64 | 256 | 16 | laplace | BN | True | 128 | 0.01 | 0.0 | 0.01 | 200 |
| **Path-X** | 4 | 64 | 128 | 32 | 128 | 16 | laplace | BN | True | 128 | 0.01 | 0.0 | 0.01 | 100 |
| **SC-Raw (base)** | 6 | 60 | 120 | 30 | 120 | 16 | laplace | BN | True | 20 | 0.01 | 0.0 | 0.01 | 200 |
| **SC-Raw (big)** | 6 | 72 | 144 | 36 | 144 | 16 | laplace | BN | True | 20 | 0.008 | 0.0 | 0.01 | 200 |

### D.3 RAW SPEECH CLASSIFICATION

Following Gu et al. (2022a), we perform speech classification on the SC10 subset of the Speech Commands dataset (Warden, 2018), which is a 10-class classification task. The chunk size of MEGA-chunk is 1000. Other hyper-parameters are listed in Table 8.

### D.4 LANGUAGE MODELING

**Training details** We use the data of WikiText-103 and enwik8 and their splits provided by Dai et al. (2019). At training time, we split the training data into segments; each segment contains $m$ consecutive chunks, where the chunk size is the effective attention length. $m$ is a random integer variable uniformly sampled from $[cl, ch]$. We use $[cl, ch] = [2, 6]$ for WikiText-103 and $[cl, ch] = [2, 4]$ for enwik8. Other training hyperparameters including optimizer, learning rate scheduler and architecture are presented in Table 9.

**Length extrapolation at inference time** We employ MEGA-chunk (§3.4) for training and set the attention chunk size to be 1024 and 2048 for WikiText-103 and enwik8 respectively. To use a longer Mega attention length at inference time than the one used at training time (i.e. 1024 or 2048), we apply rotary positional embedding (Su et al., 2021) to the attention sublayer. At test time, we split the test data into $K$ segments and sequentially process each segment by $m$ chunks, i.e. the maximum context length of each segment is $\frac{\#\text{test tokens}}{K}$. In Table 5, we report test results that use longer chunk sizes (attention lengths) of 2048 and 4096 for WikiText-103 and enwik8 respectively. MEGA can naturally extrapolate at inference time to sequences longer than those seen during training due to the recurrent design of the EMA layer. That design enables the inputs of each chunk to access the historic context through EMA as illustrated in Figure 3. On the other hand, due to the use of rotary positional embeddings, attention can be performed on longer chunk sizes at test time than those seen during training. We hope these two types of length extrapolation are clear to readers. We provide the

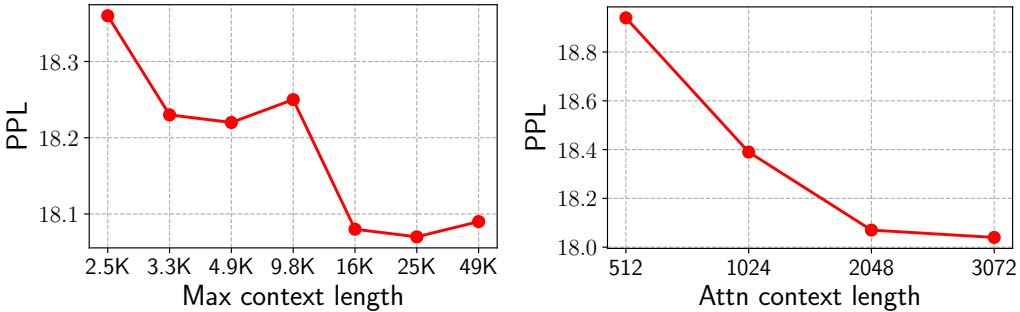

Figure 6: Ablation studies of using different context lengths and attention lengths on WikiText-103.

ablation studies on these two types of length extrapolation below, i.e. extrapolation to longer context by increasing input sequence lengths and extrapolation to longer attention lengths by increasing the chunk size.

**Ablations on context lengths** First, we fix the chunk size to be 2048 and vary $K$ within $[100, 75, 50, 25, 15, 10, 5]$ corresponding to maximum context tokens of $[2.5K, 3.3K, 4.9K, 9.8K, 16K, 25K, 49K]$. We plot the test PPL as we increase the context length in the left of Figure 6. Although at training time, the maximum context length the model has seen is 6144, MEGA can extrapolate to longer context lengths. The plot shows that PPL decreases as the context length is increased and the improvements saturate when the context length is longer than 25K. This is consistent with the observations in Press et al. (2021).

**Ablations on attention chunk sizes** Next, we fix the context length to be 25K and increase the chunk size from 512 to 3072. As shown in the right side of Figure 6, MEGA consistently improves as we increase the attention length although it only uses an attention length of 1024 during training. This contradicts with the findings in Alibi (Press et al., 2021), which finds that rotary embeddings don't generalize to longer lengths and result in higher PPL.

Table 9: Hyper-parameters of models for language modeling.

|  | WikiText-103 | enwik8 |
|---|---|---|
| Batch Size $\times$ GPUs | $6144 \times 24$ | $8192 \times 8$ |
| Optimizer | AdamW | AdamW |
| Learning Rate | 0.005 | 0.005 |
| Adam-$\beta$ | $(0.9, 0.98)$ | $(0.9, 0.98)$ |
| Learning Rate Decay | linear | linear |
| Weight Decay | 0.1 | 0.1 |
| Dropout | 0.3 | 0.1 |
| Attention Dropout | 0.1 | 0.0 |
| FFN Hidden Dropout | 0.1 | 0.0 |
| Gradient Clipping | 1.0 | 1.0 |
| Warmup steps | 24K | 24K |
| Total updates | 400K | 400K |
| Decoder Layers | 16 | 12 |
| Model size | 1024 | 512 |
| FFN Hidden size | 1536 | 1024 |
| Shared Repr. size ($z$) | 256 | 128 |
| Value Seq. size ($v$) | 1536 | 1024 |
| EMA dimension ($h$) | 16 | 16 |
| Chunk size | 1024 | 2048 |
| Total Parameters | 252M | 39M |

## D.5 MACHINE TRANSLATION

The WMT 2016 English-German dataset contains 4.5M parallel sentence pairs for training. We following the standard setting (Ott et al., 2018), using Newstest2013 as the validation set and Newstest2014 as the test set. The dataset is pre-processed following (Ma, 2020), using the scripts from FairSeq package (Ott et al., 2019).[4] We share the source and target vocabularies within the language pair, with 32K byte pair encoding (BPE) types (Sennrich et al., 2016). We report the average of both tokenized and SacreBLEU[5] (Post, 2018). The hyper-parameters of Transformer and MEGA models are listed in Table 10.

Table 10: Hyper-parameters of models for machine translation.

|  | **XFM**-Base | **MEGA**-Base |
|---|---|---|
| Batch Size $\times$ GPUs | $8192 \times 8$ | $8192 \times 8$ |
| Optimizer | AdamW | AdamW |
| Learning Rate | 0.0005 | 0.001 |
| Adam-$\beta$ | $(0.9, 0.98)$ | $(0.9, 0.98)$ |
| Learning Rate Decay | inv. sqrt | linear |
| Weight Decay | $1e-4$ | 0.05 |
| Dropout | 0.1 | 0.15 |
| Attention Dropout | 0.1 | 0.1 |
| FFN Hidden Dropout | 0.1 | 0.1 |
| Gradient Clipping | 1.0 | 1.0 |
| Label Smoothing | 0.1 | 0.1 |
| Warmup steps | 4K | 4K |
| Total updates | 500K | 500K |
| Encoder Layers | 6 | 6 |
| Decoder Layers | 6 | 6 |
| Model dimension | 512 | 512 |
| FFN Hidden dimension | 2048 | 1024 |
| Shared Repr. dimension ($z$) | – | 128 |
| Value Seq. dimension ($v$) | – | 1024 |
| EMA dimension ($h$) | – | 16 |
| Total Parameters | 65M | 67M |

---

[4]https://github.com/pytorch/fairseq
[5]signature: nrefs:1|case:mixed|eff:no|tok:13a|smooth:exp|version:1.5.1

## D.6 IMAGE CLASSIFICATION

Hyper-parameters are listed in Table 11. We closely follow Touvron et al. (2021) by reusing most of the their hyper-parameters.

Table 11: Ingredients and hyper-parameters of DeiT and MEGA.

|  | DeiT-B | MEGA |
|---|---|---|
| Batch size | 1024 | 1024 |
| Optimizer | AdamW | AdamW |
| learning rate | 0.001 | 0.002 |
| Learning rate decay | cosine | cosine |
| Weight decay | 0.05 | 0.05 |
| Epochs | 300 | 300 |
| Warmup epochs | 5 | 20 |
| Label smoothing | 0.1 | 0.1 |
| Dropout | ✗ | ✗ |
| Stoch. Depth | 0.1 | 0.2 |
| Repeated Aug | 3 | 3 |
| Gradient Clip. | ✗ | 1.0 |
| Rand Augment | 9/0.5 | 9/0.5 |
| Mixup prob. | 0.8 | 0.8 |
| Cutmix prob. | 1.0 | 1.0 |
| Erasing prob. | 0.25 | 0.25 |
| Num. Layers | 12 | 12 |
| Model size | 768 | 768 |
| FFN Hidden size | 3072 | 1536 |
| Shared Repr. size ($z$) | – | 256 |
| Value Seq. size ($v$) | – | 1536 |
| Total Parameters | 86M | 90M |

