# OpenReview forum: "Mega: Moving Average Equipped Gated Attention"
_ICLR.cc/2023/Conference — ICLR 2023 poster_

### Official Review · Reviewer_PXbY · 2022-10-18

**Confidence:** 4
**Correctness:** 4
**Technical Novelty And Significance:** 4
**Empirical Novelty And Significance:** 3
**Recommendation:** 8

**Clarity, Quality, Novelty And Reproducibility:**

* Clarity: The paper does a great job. This is all the more impressive that there are several contributions.
* Quality: This is overall a high quality paper.
* Novelty: This scores quite high, with several contributions. I could see criticism that points out that many components already exist elsewhere but I believe finding the proper combination of attention + SSM/EMA like-bias is worthwhile and I see the Laplace function as the cherry on top.
* Reproducibility: No problem here, the appendix is very helpful in this regard.

**Strength And Weaknesses:**

Strength:
- The paper does a great job at explaining the newly introduced architecture, explaining how its components differ from the transformer and how they are related to prior work. This is quite an achievement given several components are introduced.
- The key technical contributions (EMA layer for Mega, chunking for Mega-chunk) are well motivated by the issues with the classic attention block (lack of inductive bias, quadratic complexity).
- The empirical results support the strength of Mega across a wide variety of tasks. I particularly liked that Speed and Memory columns were available in table 2.
- The Laplace attention function is a nice contribution in itself and could be re-used in other work independently of the core architecture.

Weaknesses:
- More ablations on gating: While the paper does a good job on the EMA/attention function side of things, I believe some experiments could also be run on the gating mechanism. I understand that space limitations might have prevented it in this version.
- Minor ablation: How much does the method deteriorate if you do not use damping?
- S4 comparison: In section 4.1, I would point out explicitly that S4-v2 performs similarly to Mega-chunk which is the relevant baseline when accounting for speed and memory. Similarly, it would be more fair to include the Mega (base) number in Table 1 for SC-Raw, especially since the replicated S4 number is already lower than the original.

Minor spelling issues:
- TEext p8 -> Text
- Table 2 caption, remove parentheses around Long Range Arena?

**Summary Of The Paper:**

This paper introduces Mega, a gated, single-headed attention mechanism that incorporates a (damped, multi-dimensional) moving average. Instead of being computed directly on the hidden state, the query and keys are computed on a EMA-transformed version of it. The gating mechanism is akin to GRUs (with different inputs). The motivation of this approach is to introduce the EMA's inductive bias for recency/short-term relationships into the attention mechanism. Another feature of attention is its quadratic complexity in the context length, which can pose problems to model long-range dependencies. Here, the authors suggest chunking Q, K, V into k chunks and applying attention individually to each chunk to reduce it to linear complexity. This idea is not new. However because Q and K are computed on EMA-transformed inputs the attention operation can use information beyond the immediate chunk, which is in contrast with other chunking approaches. Finally, the authors also introduce the Laplace attention function as an alternative to the softmax in their attention layer.

The authors then test Mega and Mega(Chunk) on the Long Range Arena benchmark and MEGA on several long-context sequence modeling tasks (NMT, image classification, raw speech classification), finding strong performance in general. Ablations are also performed on the EMA layer, the chunk size for Mega-chunk and the attention function used.

Overall, the paper is rich in its technical contributions, combining ideas to create an architecture for long-sequence modeling that achieves strong empirical results.



**Summary Of The Review:**

This is a great paper, introducing several worthwhile technical contributions. The Mega architecture provided has interesting features and achieves strong performance across many tasks. The paper is high quality overall. There are minor criticisms, but I believe they are far outweighed by the strengths.

---

> ### Author Response · Authors · 2022-11-09
> **Response to Reviewer PXbY**
>
> Thanks for your time and constructive feedback! We appreciate your positive feedback about the technical contributions of Mega and its strong performance. We address your concerns and questions below and revise the paper to reflect the changes. Please let us know if you still have concerns after you read our response.
>
> > W1: How much does the method deteriorate if you do not use damping?
> Please see our results in the general response. The short answer is that it can consistently decrease the performance by a small margin on both text and image datasets.
>
> > W2: S4 comparisons – S4-v2 performs similarly to Mega-chunk on LRA and it would be more fair to include the Mega (base) number in Table 1 for SC-Raw.
>
> Thanks for your suggestion, and we have used Mega-base number in Table 1 for SC-Raw!

---

### Official Review · Reviewer_ZPfy · 2022-10-24

**Confidence:** 5
**Correctness:** 2
**Technical Novelty And Significance:** 2
**Empirical Novelty And Significance:** 2
**Recommendation:** 5

**Clarity, Quality, Novelty And Reproducibility:**

**Clarity:** The paper is mostly well-written, but some discussion points seem obscured. For example, in section 3.2, the authors write "the computation of the convolutional kernel in S4 requires complex fast Fourier transformers... The EMA sub-layer... convolution kernel would be a Vandermonde product, which can be computed in an efficient and numerically stable way." But EMA also uses a (real) FFT to compute it, which has the same complexity (and nearly same line in PyTorch) as a complex FFT.

**Quality:** The results are positive, but I have a few concerns.

1. Baselines.
As the authors write, the EMA layer's convolution can be computed using a Vandermonde product. This is almost exactly the same setup as S4D [1], a diagonal version of S4 (see novelty section below for more detailed comparison).
This suggests a number of natural baselines, which the authors do not evaluate:
* Adding S4D to an attention layer
* Adding S4 to an attention layer
These can be added in exactly the same way as EMA is added to attention. I would like to see these baselines to evaluate where the lift is coming from (attention, S4/EMA, or both).

2. Hyperparameter sweeps.
Based on table 8, there appears to have been extensive hyperparameter tuning for LRA. The paper is missing a description of the sets of hyperparameters swept for MEGA. I would like to see the S4+attention baselines *with the same hyperparameter sweeps* (not the exact same hyperparameters) in LRA. I also wonder whether similar hyperparameter sweeps would result in better performance for the existing baselines.

3. MEGA-chunk appears to only have been evaluated on LRA. What are the results on the other tasks?

**Novelty:** The contribution of combining EMA with attention is a good idea, and I like it.

However, the paper presents EMA as a distinct contribution from previous sequence models like S4. The paper does not discuss differences with simpler models such as S4D (which is also computed with a vandermode kernel). This is concerning since parts of the EMA code appear to be lifted directly from the S4D code base without attribution.

The forward pass is almost exactly the same as S4D's implementation, with renamed variables
https://github.com/HazyResearch/state-spaces/blob/f72202696125ce10ee7e3745ade5b04692dd2b74/src/models/s4/s4d.py#L95

S4D forward pass:
```
""" Input u shape (B, H, L) """
L = u.size(-1)

# Compute SSM Kernel
k = self.kernel(L=L) # (H L)

# Convolution
k_f = torch.fft.rfft(k, n=2*L) # (H L)
u_f = torch.fft.rfft(u, n=2*L) # (B H L)
y = torch.fft.irfft(u_f*k_f, n=2*L)[..., :L] # (B H L)
```

EMA forward pass (rename u to x, dimension H to D):
```
seq_len, bsz, embed_dim = x.size()
...
# L x B x D -> B x D x L
x = x.permute(1, 2, 0)
...
k = self.kernel(seq_len)
fft_len = seq_len
...
k_f = torch.fft.rfft(k, n=2 * fft_len)
x_f = torch.fft.rfft(x, n=2 * fft_len)
# B x D x L
out = torch.fft.irfft(x_f * k_f, n=2 * fft_len)[..., s:s + seq_len]
```

The bidirectional implementation (lines 198-204) is also the same implementation as the bidirectional implementation of S4, with renamed variables and slightly different torch operations: https://github.com/HazyResearch/state-spaces/blob/f72202696125ce10ee7e3745ade5b04692dd2b74/src/models/s4/s4.py#L1499

S4:
```
if self.bidirectional:
    k0, k1 = rearrange(k, '(s c) h l -> s c h l', s=2)
    k = F.pad(k0, (0, L)) \
            + F.pad(k1.flip(-1), (L, 0)) \
```

EMA:
```
if self.bidirectional:
    k1, k2 = torch.split(k, [self.embed_dim, self.embed_dim], dim=0)
    # D x 2*L-1
    k = F.pad(k1, (kernel_size - 1, 0)) + F.pad(k2.flip(-1), (0, kernel_size - 1))
```

The two implementations also have the same error message:

S4:
```
if state is not None:
    assert not self.bidirectional, "Bidirectional not supported with state forwarding"
```

EMA (rename "state forwarding" to "incremental state"):
```
assert not self.bidirectional or incremental_state is None, 'Bidirectional EMA does not support incremental state'
```

**Reproducibility:** The code and final hyperparameters have been provided, but the hyperparameter sweeps are missing.

**Misc:** The code is not properly anonimized, it contains references to the "fairseq" library.

[1] Gu et al. On the the Parameterization and Initialization of Diagonal State Space Models. https://arxiv.org/abs/2206.11893

**Strength And Weaknesses:**

+ Improvement over S4 on a number of tasks, including language modeling.
+ Extensive experiments and hyperparameter tuning to show that this method has promise.

- EMA appears to be almost identical to S4D, a diagonal version of S4, but the authors do not compare against simple baselines such as replacing EMA with S4 or S4D. Thus it is unclear where the benefit comes from.
- Some claims about the benefits of EMA over S4 are confusing.
- The code implementation of EMA seems lifted from S4D [1], with no discussion of the differences or benefits, or acknowledgment of S4D. This should be corrected in the paper.

[1] Gu et al. On the the Parameterization and Initialization of Diagonal State Space Models. https://arxiv.org/abs/2206.11893

**Summary Of The Paper:**

This paper proposes MEGA, a method for combining exponential moving average (EMA) with attention. The paper evaluates MEGA on a number of benchmark datasets for sequence modeling and finds positive results.

**Summary Of The Review:**

Overall, this paper introduces an interesting idea that deserves more exploration. However, I have some concerns about the novelty, the merits of the evaluation, and the framing.

---

> ### Author Response · Authors · 2022-11-09
> **Response to Reviewer ZPfy**
>
> Thanks for your time and constructive comments! We appreciate your positive feedback on the good motivation and promising results. We respond below to your questions and comments. We kindly request that you briefly read these responses and let us know if they do not fully address your concerns.
>
> >W1: Presentation of EMA and comparison with S4(D).
>
> We appreciate your suggestion to clearly explain the relation between EMA and state space models. We add more descriptions to explain the connections between EMA and S4D in our revised version. In the paper, we presented EMA in the view of a moving average approach as it was commonly used as a smoothing method in statistics, not in the view of state space models. Since the state space model (SSM) is a generic framework, S4, S4D and EMA can be considered as specific instantiations of SSM.  In fact, in the last paragraph of section 3.2, we discussed the relationship and difference between EMA and S4, clearly saying that EMA can be seen as a simplified variant of SSM with a diagonal state matrix. Moreover, EMA in Mega restricts the diagonal state matrix to be real and the elements in the range of (0, 1), and does not rely on the HiPPO framework for parameter initialization. S4D is a concurrent work which also applies diagonalization on the state matrix, but the elements in its state matrix are **complex with no restrictions**. In addition, **S4D still relies on the HiPPO framework for parameter initialization**. The code you displayed in your review is using FFT to compute the output given the precomputed kernel. This part (and the code) is exactly the same between S4, S4D and EMA. The difference is the computation of the kernel. In addition, S4D was not published when we implemented Mega and we didn’t borrow any code from it.
>
> > W2: Some claims about the benefits of EMA over S4 are confusing. in section 3.2, the authors write "the computation of the convolutional kernel in S4 requires complex fast Fourier transformers... The EMA sub-layer... convolution kernel would be a Vandermonde product, which can be computed in an efficient and numerically stable way." But EMA also uses a (real) FFT to compute it, which has the same complexity (and nearly same line in PyTorch) as a complex FFT.
>
> Yes, both EMA and S4 need FFT to compute the final output **given their kernels**. The difference is that **S4 even requires complex FFT to compute its kernel** (see Alg 1 in [1]), while EMA only needs a Vandermonde product.
>
> >W3: Ablation study by replacing EMA with S4 and S4D.
>
> We appreciate your suggestion of this ablation study and have provided the results in the general response.
>
> >W4: Hyperparameter sweeps. Based on table 8, there appears to have been extensive hyperparameter tuning for LRA. The paper is missing a description of the sets of hyperparameters swept for MEGA. I would like to see the S4+attention baselines with the same hyperparameter sweeps (not the exact same hyperparameters) in LRA. I also wonder whether similar hyperparameter sweeps would result in better performance for the existing baselines.
>
> We appreciate your suggestion. As stated in the second paragraph of section 4.1, we set most of the hyperparameters by following S4-v1[1], with minor adjustments of model dimensions for fair comparison. In fact, S4-v2 [2] used larger model sizes and better-tuned hyperparameters (e.g. different learning rates and weight decay rates for different layers) than S4-v1. But due to the limitations of time and computational resources, we did not enlarge Mega to match the model sizes in S4-v2 and further tune hyperparameters. With that being said, S4-v2 is more heavily tuned than Mega.
>
> >W5: MEGA-chunk appears to only have been evaluated on LRA. What are the results on the other tasks?
>
> We apologize for the confusion. In fact, we also evaluated Mega-chunk on language modeling and raw speech classification. For example, in raw speech classification we use Mege-chunk with chunk size 1000 to classify raw speech with length 16,000. We will revise the paper to make it clearer.
>
> > Q1: fairseq is included in the submitted code.
>
> To our best knowledge, fairseq is a publicly open sourced package so we think this does not violate anonymity.
>
> References:
> [1] Gu et al., 2022. Efficiently Modeling Long Sequences with Structured State Spaces.
> [2] Gu et al., 2022. On the parameterization and initialization of diagonal state space models.

---

> > ### Comment · Reviewer_ZPfy · 2022-11-11
> > **Response**
> >
> > Thank you for your response.
> >
> > Given the similarities between EMA and S4D (both diagonal, both use Vandermonde product to compute the kernel), I think it would be helpful to have a more upfront discussion about the differences. As Albert writes in his public comment, the major differences appear to be that S4D allows complex parameters, while EMA only allows real parameters, and EMA has a simpler initialization. I think it would strengthen the paper to acknowledge and discuss this connection up front, and not leave it to one line on page 5.
> >
> > Another question -- what hardware did you use to make the speed and memory measurements for LRA? I'm curious about the memory requirements for PathX, since I believe the attention component should still require quadratic memory footprint. With batch size 128, this seems larger than would fit on a single GPU.

---

> > > ### Author Response · Authors · 2022-11-12
> > > **Response**
> > >
> > > Thanks for your response!
> > >
> > > > Given the similarities between EMA and S4D (both diagonal, both use Vandermonde product to compute the kernel), I think it would be helpful to have a more upfront discussion about the differences. As Albert writes in his public comment, the major differences appear to be that S4D allows complex parameters, while EMA only allows real parameters, and EMA has a simpler initialization. I think it would strengthen the paper to acknowledge and discuss this connection up front, and not leave it to one line on page 5.
> > >
> > > We just updated our paper and describe these connection between EMA and S4D in the last paragraph of Section 3.2 by explicitly displaying the three differences:
> > > 1. S4D parameterizes state matrix in the complex field while EMA uses real numbers.
> > > 2. EMA restricts the diagonal elements in the range of $(0, 1)$.
> > > 3. EMA does not rely on the HiPPO framework for parameter initialization.
> > >
> > > In fact, S4D is a concurrent work of Mega. We have completed the design and implementation of Mega before S4D was released. We hope this revision satisfies your requirement.
> > >
> > > > Another question -- what hardware did you use to make the speed and memory measurements for LRA? I'm curious about the memory requirements for PathX, since I believe the attention component should still require quadratic memory footprint. With batch size 128, this seems larger than would fit on a single GPU.
> > >
> > > All the LRA experiments were conducted on a single Nvidia A100 GPU with 40GB of memory, except Path-X ones which were  on 8 x A100 GPUs.
> > >
> > > Please let us know if we have addressed your concerns, thanks!

---

> > > > ### Comment · Reviewer_ZPfy · 2022-11-12
> > > > **Response**
> > > >
> > > > Thank you, the connection is clearer now.
> > > >
> > > > For Path-X - how do you compute the speedup and memory savings numbers?

---

> > > > > ### Author Response · Authors · 2022-11-12
> > > > > **Response**
> > > > >
> > > > > > For Path-X - how do you compute the speedup and memory savings numbers?
> > > > >
> > > > > As described in Section 4.1, we followed previous works and examined the speed and memory efficiency of MEGA on the ***byte-level classification task*** with the input length of ***4K***. So we did not evaluate the efficiency of Mega on Path-X.

---

> > > > > > ### Comment · Reviewer_ZPfy · 2022-11-12
> > > > > > **Response**
> > > > > >
> > > > > > Thank you for the clarification, I missed that detail.
> > > > > >
> > > > > > Looking forward to seeing the rest of the ablations!

---

> ### Author Response · Authors · 2022-11-17
> **Have we addressed your concerns?**
>
> Thanks for your reviews again! We have revised the paper to better illustrate the connections between EMA and S4D, and we have added additional experiments to compare EMA and state space model variants in the [general response](https://openreview.net/forum?id=qNLe3iq2El&noteId=lIE25xPHz70). Would you please reconsider the ratings of our paper? Please let us know if you still have any concerns regarding our work, thank you!

---

> > ### Comment · Reviewer_ZPfy · 2022-11-17
> > **Response**
> >
> > Hi, I still have two concerns:
> >
> > **Positioning**
> > I think the paper would be better positioned with a discussion of the connection between EMA and S4/S4D up front (as early as the intro). The detailed comparison on page 5 is clear, but the paper would be stronger with better upfront framing.
> >
> > For example, if this paper is instead positioned as a method for combining state space models with attention, then I think it properly acknowledges the previous work - and your intriguing results about how the differences between how EMA and S4D play out in MEGA (vs when they are used on their own) becomes an interesting ablation study.
> >
> > As it stands, it appears that you can swap out EMA for multiple other SSM representations, and the results are roughly the same. If the positioning is "here is a way to combine SSMs with attention" - then this is a **strength** (it means that your method for combining SSMs and attention is robust to choice of SSM). As it stands, the paper seems to claim that EMA is a better choice than other SSM formulations. Based on how close your ablations are, and the apparent difficulty in reproducing them, this claim seems shaky.
> >
> > **Hyperparameters**
> > I am also still concerned about how the hyperparameters are tuned. As you write in your response to Albert's attempted reproduction of your results, it appears that some of the methods have significant degradations in quality based on hyperparameters like the number of warmup steps. It would be helpful to know how the hyperparameters were tuned (what the sweeps were) for reproducibility.

---

> > > ### Author Response · Authors · 2022-11-17
> > > **Response**
> > >
> > > > **Positioning** I think the paper would be better positioned with a discussion of the connection between EMA and S4/S4D up front (as early as the intro). The detailed comparison on page 5 is clear, but the paper would be stronger with better upfront framing.
> > >
> > > As we claimed in our paper and replied in our previous response, the main motivation of Mega is to incorporate stronger inductive bias by the classic EMA approach, **not** a method for combining state space models with attention. Since the state space model (SSM) is a generic framework, S4, S4D and EMA can be considered as specific instantiations of SSM. We have discussed the connections between them. As our paper is **not** positioned as a combination of attention and SSM, we believe that the ablation study of replacing EMA with other SSM formulations is not the central contribution. In addition, we are not claiming that EMA is a better choice of SSM than S4 and/or S4D, but finding the optimal SSM formulation in Mega is out of the scope of our paper.
> > >
> > > > **Hyperparameters** I am also still concerned about how the hyperparameters are tuned. As you write in your response to Albert's attempted reproduction of your results, it appears that some of the methods have significant degradations in quality based on hyperparameters like the number of warmup steps. It would be helpful to know how the hyperparameters were tuned (what the sweeps were) for reproducibility.
> > >
> > > As we replied in the previous response, we set most of the hyperparameters by following S4-v1, with minor adjustments of model dimensions for fair comparison. We tuned some key hyper-parameters based on training convergence and validation accuracy, such as `learning rate`, `weight decay` and `warmup steps`, but it does not mean that Mega is more heavily tuned than other models. In fact, tuning `warmup steps` is very common in attention-based models, because attention is somehow unstable at the beginning stage of training, and it is important to set a proper and long enough warmup period. In addition, some hyper-parameters vary a lot for different model architectures and tasks. For example, in Transformer for NLP tasks, people usually set `beta2=0.98` in the AdamW optimizer instead of the default value `0.999`. Thus, it is important and necessary to tune hyper-parameters properly for different models and tasks, to guarantee  that all the models are running in their optimal modes for fair comparison.

---

### Official Review · Reviewer_nvWf · 2022-11-01

**Confidence:** 4
**Correctness:** 3
**Technical Novelty And Significance:** 3
**Empirical Novelty And Significance:** 3
**Recommendation:** 8

**Clarity, Quality, Novelty And Reproducibility:**

The paper is well structured from formal definition, illustrations, experimental result, and analysis. All figures are helpful to understand the architecture and computational graph of MEGA.

Since large pre-trained models are most widely used, I wonder the effectiveness of MEGA can be also generalized to those models. In my understanding, EMA only calculates new representations based on the previous time positions. For example, BERT brought huge success from attending to tokens in both directions. In other words, I suspect EMA component may harms a gain from bidirectional interactions.

I couldn’t check Appendix B thoroughly but theoretical justification part (Secion 3.5) is very nice.

MEGA-chunk achieves good speed gain. However, it introduces additional hyperparameter (chuncking size) depending on different tasks. A method (or module) that can automatically decide this value might be needed.


**Strength And Weaknesses:**

MEGA shows superior performance on diverse datasets, meaning its practical usefulness. The motivation of MEGA is well supported and the solution is not too much complex. However, there are still many not straightforward design choices so more ablations on them could improve the completeness of the paper. It would be great if the authors include a comparison table about computational complexity in terms of FLOPs, latency, the number of parameters, memory footprint, etc. Also, having results on different scale might give a new insight.


**Summary Of The Paper:**

This paper propose moving average equipped gated attention mechanism (MEGA) to address  transformer’s weakness in long-range modeling such as weak inductive bias and quadratic computational complexity based on the idea of EMA (more specifically, multi-dimensional damped EMA). MEGA (and its efficient variant, MEGA-chunk) can replace multi-head attention part of the original transformer. MEGA outperforms all other baselines including vanilla transformer and S4 on various benchmarks including Long Range Arena and different modalities. MEGA-chunk underperforms MEGA but it is much efficient.

**Summary Of The Review:**

Overall, the paper is well written with principled design and good empirical results. Although the method can be regarded as a combination of existing methods, I believe MEGA can be highly useful considering the impact of commonly used architectures like transformer. Nevertheless, more rigorous evaluation on more diverse scenario might be necessary.

---

> ### Author Response · Authors · 2022-11-09
> **Response to Reviewer nvWf**
>
> Thanks for your time and constructive feedback! We appreciate your positive feedback about the principled design of Mega and its strong empirical performance. We address your concerns and questions below and revise the paper to reflect the changes. Please let us know if you still have concerns after you read our response.
>
> > W1:  I wonder if the effectiveness of MEGA can be also generalized to large-scale pre–training models. I suspect EMA component may harms a gain from bidirectional interactions.
>
> First, we apologize for the unclarity of the usage of EMA described in the paper and have revised it. EMA can be used in a bi-directional manner and indeed we used the bidirectional EMA for all non-autoregressive modules (encoders) throughout our experiments, although the mathematical formula of EMA presented in the paper is in the unidirectional form. The extension of using bidirectional EMA is straightforward where we simply have two sets of EMA parameters, one for each direction and we execute EMA from both directions and add their representations. Second, we haven’t got our hands full with the scaling experiments of Mega yet but this is definitely an exciting direction we are planning to work on.
>
> > W2: chunk size is an additional hyperparameter depending on different tasks. A method (or module) that can automatically decide this value might be needed.
>
> Thanks for your suggestions and we think this is an interesting idea! In our experiments, we didn’t tune the chunk size as a hyperparameter for all the experiments that used Mega-chunk, instead, we chose a reasonable chunk size for different tasks, e.g. we use a chunk size of 128 for all LRA tasks except Pathfinder-X and we use a chunk size of 1024 or 2048 for LM datasets.
>
> > Q1: It would be great if the authors include a comparison table about computational complexity in terms of FLOPs, latency, the number of parameters, memory footprint, etc.
>
> Thanks for your advice! We have reported the number of parameters for all tasks. In the LRA experiments, we compared the training speed and memory footprint between Mega and baseline methods. For the language modeling task, we have compared the inference latency between Mega and the Transformer model. We will try to include a more complete picture of computational complexity in the camera-ready version.

---

### Official Review · Reviewer_Dc4o · 2022-11-01

**Confidence:** 3
**Correctness:** 3
**Technical Novelty And Significance:** 3
**Empirical Novelty And Significance:** 3
**Recommendation:** 8

**Clarity, Quality, Novelty And Reproducibility:**

**Novelty/Originality**: Although aspects of these work have been presented before (for example SRU++ have combined recurrency and attention in a similar, but not same, manner. Chunking based attention isn't new. EMA is a old technique which is combined with strategies used in S4-like models -- leading to a simpler form of SSM. Adding GRU/LSTM style gating in between layers in Transformers have been done before too), the novelty is still significant (theoretical connection between gated single headed attention and multi headed attention is interesting, Multidimensional dampled EMA seems novel despite being a mixture of prior works, the setup is also more effective than SRU++ going by footnote 2, mixing effective techniques together (even if none are individually new) to show good empirical performance is still a significant contribution and so on)


**Clarity**: Good.

**Quality**: High.

**Reporducibility**: Decent. I am a bit unclear about the positional encoding being used in different tasks, but otherwise most hyperparameters are clarified and the model code is also released.


**Questions**

1) Just to confirm if I am understanding correctly: non-chuncked MEGA still uses the nxn quadratic attention but nevertheless gets memory performance similar to BigBird. Is it because of having more EMA parts and single headed attentions?

2) What positional encoding are used in different tasks? Is it ROPE for all?

3) Can you address this concern:

I had some concerns with Theorem 1.

It seems that gamma is a function of single token representation. While the result to be recovered is a function of the whole sequence. So it's not clear if in theory gated single headed attention is indeed capable of approximating multi-headed attention unless I am misunderstanding something.

Particularly, say we have a sequence of hidden states:

$H = (h_1, h_2, \dots, h_n)$

Now,

$\gamma^i_t = U(h_t)$ where $U$ is some universal approximator (eqn 12), $\gamma_t$ is the gate at the $t^{th}$ position, $i$ indicates the head.

And for single headed attention at position t can be formulated as:

$\tilde{h}_t = SHA(H, t)$

where $SHA(H, t) = \sum_{i=1}^n \alpha_{t,i} f_v(h_i)$

( $f_v$ is a function that makes value transformation; typically a linear layer, and $\alpha_{t,i}$ is the attention coefficient)

Similarly, we may have a multiheaded attention (MHA) based representation at position $t$, for head $i$.

$\hat{h}^i_t = MHA(H, t)$

The idea is to approximate MHA with gated SHA like this:

$\tilde{h}_t \odot \gamma^i_t = \hat{h}^i_t$

That is, we want:

$SHA(H,t) \odot \gamma^i_t = MHA(H,t)$

$\implies  \gamma^i_t = MHA(H,t) \oslash SHA(H,t)$

Let's say $R(H,t) = MHA(H,t) \oslash SHA(H,t)$ (abbreviation)

The claim in the paper seems to be that since $\gamma^i_t$ is the result of an universal approximating function, it can theoretically recover $R(H,t)$.

However, there seems to be a mistake. Since strictly speaking $\gamma^i_t$ only relies only on the information at position t ($h_t$) whereas $R()$ is a function of the whole sequence of hidden states even for the output representation at position i.

More formally, since, we defined $\gamma^i_t = U(h_t)$, the claim in the paper boils down to saying:

$U(h_t) = R(H,t)$

That is, the claim boils down to saying that we can recover  $R(H,t)$ just from $U(h_t)$. But even if $U(.)$ is a universal approximator its true input differs from the input of the function it is trying to approximate. So I am not sure how we can get a guarantee for equivalence in theoretical capacity for Gated SHA and MHA.


We can try to address this issue by removing the temporally-indexed formalism as:

$\gamma^i = \tilde{U}(H)$ where

$\tilde{U}(H) = (U(h_1), U(h_2), ...., U(h_n)) = (\gamma^i_1, \gamma^i_2, ...., \gamma^i_n)$

But now even if $U(h_t)$ is a universal approximator of any continuous function $f(h_t)$ by virtue of $U()$ being a non-linear wide neural network over $h_t$, $\tilde{U}(H)$ can't be said to be a universal approximator of any function $f(H)$. This is because $\tilde{U}(H)$ is not simply a neural network over some vector $H$ but it is a position-wise repeated application of the neural network $U()$ over each token representations in the sequence $H$.

**Strength And Weaknesses:**

Strength:

1. Effective motivated mixture of multiple prior techniques. Introduction of better activation functions, expansion of EMA in a multidimensional form similar to S4.

2. Strong empirical performance accross multiple modalities and tasks (speech classification, imagenet, LRA, language modeling, translation).

3. Several relevant ablations are present (for example, removing EMA, or changing dimensions of EMA and so on).


Weaknesses:


1. If I understand correctly you are using using a unidirectional EMA (going from left to right). If so the tokens wouldn't necessarily have context from the right side. That may counter the justification that EMA mitigates the issues of chunking (it may mitigate for the left boundary of the chunk, but the right boundary will still remain unable to access the further right side context).

2. I also have some concerns with the theoretical proof in Theorem 1 for the equivalence of gated SHA and MHA. See the next section for details.

Not necessarily strong weakness but additional ablations and experiments that could have been nice:

3. I would be curious for some additional experiments/investigation. For example, what if *only* the EMA is used along with FFNs like in a S4 sort of setup (adjusted accordingly to keep parameter count similar)? Can it outperform or get close to S4 in LRA?

4. Similar to 2. but what if S4 or other existing variants (instead of EMA) are used with attention. Can it offer even better performance?

5. While thereotically, the gating function has the potentially to recover the "missing" portion from multiheaded attention, that may not always translate to similar empirical performance. I wonder if there could be some empirical performance loss from using single-head gating vs multi-headed attention. (**retracted** since there are prior works showing empirical strength of gated single-headed attention)

6. I am unclear about which positional encoding is being used in all the tasks. I understand that ROPE was used in language modelling but is it used everywhere else too? I was also wondering for a possible ablation to see if a positional encoding is even as much necessary anymore given that Multidimensional EMA may model some positional aspects already.



**Summary Of The Paper:**

* The paper adds better inductive bias to Transformer attention by using a linear recurrence in the form of exponential moving average to contextualize the queries and keys.

* The paper improves simple damped EMA into a multidimensional damped EMA turning it effectively in a simplified form of S4 where a diagonal bounded weight matrix is being used instead of HiPPOs.

* The multidiemsnional EMA is effectively integrated with Transformer stype attention through the prior framework of GAU and GRU based gating.

* The attention function is (computationally) simplified from multiheaded attention to gated single headed attention which is shown to be theoretically as expressive. A new activation function is introduced too for attention.

*  The above component forms the MEGA block. The attention can be also chunked leading to MEGA-chunk. MEGA/MEGA-chunk can then be normed and put through FFN blocks.

* Multiple experiments and ablations across various tasks in different modalities show the promise of the model.

**Sorry for mid-rebuttal update**: I just had some thoughts that raise some concerns about some of the claims (about Theorem 1) made in the paper. I wanted to put it here as soon as possible so that the authors can engage with it. I have reduced my original score (8) to (5) due to the concerns. I am willing to improve the score again once that's resolved.

**Summary Of The Review:**

The paper combines multiple prior techiques and extends several of them with novel elements and shows strong performance among multiple domains and tasks. There are some additional experiments I would be curious about and there are some unclearity about positional encodings, and concerns about the theoretical proofs but otherwise it's a solid paper.

---

> ### Author Response · Authors · 2022-11-08
> **Response to the concern of Theorem 1**
>
> We really appreciate your comments. We use this thread to address your concern on Theorem 1, and will address your other comments in another thread later.
>
> Following your notations, we use $H = (h_1, h_2, \ldots, h_n)$ as the input embedding sequence of one attention layer.
>
> The most important point which raises your concern about Theorem 1 is that at each time step $t$, the gate only takes $h_t$ as input, i.e.  $\gamma_t = U(h_t)$.
> However, in our Mega attention layer, the gate $\gamma_t$ is using the output of the EMA layer instead of directly using $h_t$ as input (please see Eq (6) and Eq (12)).
>
> Concretely, suppose we denote the output of EMA as $\hat{H} = \mathcal{E}(H)$, where each $\hat{h}_t = \mathcal{E}(H, t)$ uses the whole sequence $H$ as input. Then, $\gamma_t = U(\hat{h_t}) = U(\mathcal{E}(H, t))$ and we can assume $\gamma_t$ as a universal approximator of $H$.
>
> In addition, even in a pure gated attention layer without EMA sub-layer, we can still address this issue by assuming that there are sufficient contextual information in $H$. Your concern is based on assuming that the representations at different time steps $h_t$ are independent with no contextual information. However, in practice, $H$ is usually the output from the previous attention layer, where each $h_t$ has access to the information from other time steps. Therefore, $\gamma_t$ can theoretically recover $R(H, t)$.
>
> Hope this has addressed your concern.
>
> For your following question related to Theorem 1
>
> >Q: While thereotically, the gating function has the potentially to recover the "missing" portion from multiheaded attention, that may not always translate to similar empirical performance. I wonder if there could be some empirical performance loss from using single-head gating vs multi-headed attention.
>
> For your question about the gap between theoretical justification and empirical performance of single-head gated attention, previous work [1,2] have performed comprehensive comparison and concluded that single-head gated attention is as performant as multi-head one.
>
> [1] Hanxiao Liu, Zihang Dai, David So, and Quoc V Le. Pay attention to MLPs. Liu et al. Advances in Neural Information Processing Systems, 34:9204–9215, 2021.
>
> [2] Weizhe Hua, Zihang Dai, Hanxiao Liu, and Quoc Le. Transformer quality in linear time. In International Conference on Machine Learning (ICML-2022), pages 9099–9117. PMLR, 2022.

---

> > ### Comment · Reviewer_Dc4o · 2022-11-09
> > **Still some concerns**
> >
> > I still have some concerns regarding theorem 1:
> >
> > I was already treating $H$ as output of EMA, but even with your notation, it won't be $R(H,t)$ anymore but $R(\hat{H},t)$
> >
> > Let me be more precise. More accurately, I think $MHA()$, and $SHA()$ should be function of both $H$ and $\hat{H}$ (if we put eqn. 7 inside attention block for the convenience of the formalism) because following eqn 8 and 9, Q and K depend on Z which depends on $\hat{H}$. But you can resolve that by consider some functions like $MHAex(H, t) = MHA(\mathcal{E}(H),H, t)$ and same for $SHAex(H,t)$, and then expressing $R(H,t) = MHAex(H, t) \oslash SHAex(H,t)$. The math may need to be in the appendix, in this case.
> >
> > But even after that there still seems to be an issue.
> >
> > What we still need to show is that:
> >
> > $U(\mathcal{E}(H, t))$ can recover $R(H,t)$.
> >
> > However even if $U(.)$ is an universal approximator, it isn't true that $U \circ \mathcal{E}$ is a universal approximator for any arbitrary function $\mathcal{E}$.
> >
> > To see why, we can consider a counter example:
> >
> > Assume we want to apporximate a +1 function: $plus(x) = x+1$. Assume $U$ is a universal approximator and can recover $plus(x)$ ($U$ can learn to be $U(x) = plus(x) = x+1$)
> >
> > Assume $\mathcal{E}$ is a "zero out function" (multiplies whatever by 0) ($\mathcal{E}(H, t) = 0$)
> >
> > Now we can see, even if $U(x)$ can recover $plus(x)$, having this adversarial $\mathcal{E}$ function $U \circ \mathcal{E}(x)$ cannot implement x+1, because $\mathcal{E}$ can erase $x$. $U$ is still a universal approximator for its immediate input, but $x$ isn't its immediate input. Thus $U \circ \mathcal{E}(x)$ cannot universally approximate any arbitrary continuous function $f(x)$ for any arbitrary $\mathcal{E}$.
> >
> > Now, to be sure $\mathcal{E}$, here, isn't just any arbitrary function but EMA but my concern is: I am not sure if the universal approximation of $U(x)$ is broken or not in $U \circ \mathcal{E}(x)$ in case $\mathcal{E}$ is EMA.
> >
> >
> > In a sense, $EMA()$ as implemented here is already somewhat adversarial. Since it's unidirectional, at any timestep t, $\hat{h}_t$ would have no information related to t+j positions (for positive j), whereas the attention functions can utilize information from both direction. So it would again seem to break universality. You can still appeal to the fact that in the in-between layers, there would be earlier layers of attentions and if those layers create a good enough contextualized representation to keep all the relevant information, in practice the multi-headed attention values can be recovered. But then, I feel like the mathematical point becomes "weak", because a lot of burden for making single headed gated attention to work is moved away from it but to earlier layers to do a good job in making contextual representations (and also, the equivalence wouldn't hold strictly anymore = for example, when we have single layer pure gated SHA vs MHA without any ema, or prior contextualization).
> >
> >
> > Note: To be clear, I am fine with accepting the paper even if the math doesn't hold  (since you can still simply appeal to empirical justification from prior work). But in case if the math doesn't hold, I wouldn't be willing to raising the score until the framing in the paper is cleaned up. And it would be also good to add relevant citations and acknowledgement pointed out by $ZPfy$ (particularly, before section 3.3 you could mention S4D and how multidimensional EMA turns out to be a variant of S4D and cite it in that context; although I realize the works may be a bit too concurrent).
> >
> > ---------------
> > > For your question about the gap between theoretical justification and empirical performance of single-head gated attention, previous work [1,2] have performed comprehensive comparison and concluded that single-head gated attention is as performant as multi-head one.
> >
> > > [1] Hanxiao Liu, Zihang Dai, David So, and Quoc V Le. Pay attention to MLPs. Liu et al. Advances in Neural Information Processing Systems, 34:9204–9215, 2021.
> >
> > > [2] Weizhe Hua, Zihang Dai, Hanxiao Liu, and Quoc Le. Transformer quality in linear time. In International Conference on Machine Learning (ICML-2022), pages 9099–9117. PMLR, 2022.
> >
> > Thank you for the clarification. I will retract that point.

---

> > > ### Author Response · Authors · 2022-11-09
> > > **Response to concern on Theorem 1.**
> > >
> > > Thanks for your quick response.
> > >
> > > First, we apologize for the unclarity of the usage of EMA described in the paper. EMA can be used in a bi-directional manner and **in fact we used the bidirectional EMA for all non-autoregressive modules (encoders) throughout our experiments**. Thus, $\hat{h}_t$ contains all necessary information.
> > >
> > > We really appreciate your comments on the concern about the universal approximation of $U\circ \mathcal{E}$. Here we do not want to argue if EMA on $H$ would break the universality of approximation of $U$ because it is hard to prove. What we want to stress is that **the mathematical point of Theorem 1 would not become weaker even if we assume contextual information in $H$**.
> > >
> > > Please see Eq 19 in the paper. From Eq 19, we see that single head attention $SHA()$ is strictly "weaker"  than $MHA()$ even if we have the same assumption on $H$ (sufficient contextual information in $H$), because the one set of attention weights $a$ in $SHA()$ is "impossible" to approximate the $h$ sets of attention weights $a^1, a^2, \ldots a^h$ in $MHA()$. The purpose of Theorem 1 is to show that by adding a gate $\gamma$, it is "possible" for $SHGA()$ to approximate $MHA()$.
> > >
> > > Hope this has addressed your concern ;-)
> > >
> > > >And it would be also good to add relevant citations and acknowledgement pointed out by  (particularly, before section 3.3 you could mention S4D and how multidimensional EMA turns out to be a variant of S4D and cite it in that context; although I realize the works may be a bit too concurrent).
> > >
> > > Yes, we are revising the paper to reflect these comments and will upload the revised version soon.

---

> > > > ### Comment · Reviewer_Dc4o · 2022-11-09
> > > > **On clarity**
> > > >
> > > > It somewhat addresses my concerns but I think the theorem needs to be clearer on these points because it feels a bit misleading:
> > > >
> > > > * The theorem refers to works on universal approximation thoerems about standard wide neural network to justify the potential of $\gamma$
> > > >
> > > > * However your actual argument relies on EMA to provide sufficient contextualization. This weakens the rigor of the proof since it relies on the universality of $U$ (when referring to the literature) not $U \circ \mathcal{E}$.
> > > >
> > > > * Another problem (or rather somewhat of a "lite tension" that arises) is that it kind of shifts the burden of approximating part of the multi-head attention ability to EMA. This however runs counter to the original motivation of attention function and EMA having different inductive biases. (I understand it's strictly not a mathematical concern because theoretical capability in principle is one thing, and inductive biases are another. But the "tension" arises in the fact that on one hand we are justifying the use of attention for its distinct inductive bias (or lack thereof) but on the other hand we are implicitly assuming that EMA can do sufficient contextualization for $U$ to approximate the multi-headedness despite that EMA may lack similar kind of inductive biases to justify ignoring multi-headedness a priori)
> > > >
> > > > * Anyway, I am not saying this is really a "strong" issue so to say. At least, however, I think it would be good to make some edits (particularly noting reliance on the EMA for the gating to have the potential for approximation, and making the assumptions involving universality of $U \circ \mathcal{E}$ more explicit) so that similar confusions don't arise.
> > > >
> > > > ---------------------------
> > > >
> > > > I have increased my score again for your experimental efforts and addressing most of the other concerns.

---

> > > > > ### Author Response · Authors · 2022-11-09
> > > > > **Re: On clarity**
> > > > >
> > > > > We really appreciate your comments on improving the clarity of Theorem 1 and will revise the paper later.
> > > > >
> > > > > And we are happy to see that our response and new results have addressed your other concerns :-)

---

> ### Author Response · Authors · 2022-11-09
> **Response to Reviewer Dc4o**
>
> Thanks for your time and constructive feedback! We appreciate your positive feedback about the novelty of Mega and its strong empirical performance. We address your concerns and questions below and revise the paper to reflect the changes. Please let us know if you still have concerns after you read our response.
>
> > W1: Is EMA unidirectional and would that weaken the expressiveness of EMA to model the context?
>
> We apologize for the unclarity of the usage of EMA described in the paper and have revised it. The answer to this question is no. The mathematical formula of EMA we presented in the paper is in the unidirectional form, however, we use bidirectional EMA for all non-autoregressive modules (i.e. encoders)  in all our experiments. The extension of using bidirectional EMA is straightforward where we simply have two sets of EMA parameters, one for each direction and we execute EMA from both directions and add their representations.
>
> > W2: Concerns with Theorem 1:
>
> We respond to this in a separate thread.
>
> Next we address your questions and suggestions below:
>
> > Q1: What positional encoding are used in different tasks? Is it ROPE for all?
>
> We only used ROPE for the language modeling tasks but not for other tasks. For other tasks, we simply use very simple learnable relative biases, i.e. one learnable parameter for each relative positional distance (e.g. -3, +4). The reason why we use ROPE for LM is because we conducted experiments of length extrapolation to longer attention chunks at inference time (Appendix D4.). We clarified the use of positional embeddings in the appendix (D1.) of the revised version.
>
> > Q2: non-chuncked MEGA still uses the nxn quadratic attention but nevertheless gets memory performance similar to BigBird. Is it because of having more EMA parts and single headed attentions?
>
> Non-chunked Mega uses similar memory to BigBird for long input sequences mainly because we use single head attention. Since we tested on input sequences with length 4K, the computation of attention matrix is the bottleneck and reducing to single head from 8 heads significantly reduces the computational costs.
>
>
> > Q3: Ablation (1) – Replacing EMA with other variants of S4.
>
> Please see the results in our general response. Short answer is that we don’t see improvements and replacing EMA with S4 leads to unstable training.
>
> > Q4:  Ablation (2) — Removing relative positional encodings as EMA already encodes the positional aspects.
>
> Please see the results in our general response. Short answer is that we see a significant drop in the performance of image tasks but a slight decrease in the text tasks.
>
> > Q5: Ablation (3) —- Replacing the S4 sublayer with EMA in the S4 setup.
>
> We are working on this experiment now and the results have not been available yet. It is because we need to integrate our EMA layer into the S4 architecture and further tune hyper-parameters for fair comparison. We will update the results soon.
>
> > Q6: I wonder if there could be some empirical performance loss from using single-head gating vs multi-headed attention.
>
> Our single-head gated attention is similar to GAU [1], and in their ablation studies (Table 1) they have shown that replacing single head attention with multi-head attention does not bring any performance gains. Our preliminary results also suggest the same conclusion.
>
> > Q7: how multidimensional EMA turns out to be a variant of S4D
>
> In the original last paragraph of section 3.2, we discussed the relationship and difference between EMA and S4, saying that EMA can be seen as a simplified variant of SSM with a diagonal state matrix. Moreover, EMA in Mega restricts the diagonal state matrix to be real and the elements in the range of (0, 1), and does not rely on the HiPPO framework for parameter initialization. S4D is a concurrent work which also applies diagonalization on the state matrix, but the elements in its state matrix are **complex with no restrictions**. In addition, **S4D still relies on the HiPPO framework for parameter initialization**. We have included these discussions in the last paragraph of Section 3.2.
>
> [1] Transformer Quality in Linear Time, Hua et al., ICML 2022

---

### Public Comment · ~David_W._Romero1 · 2022-11-08
**Wrt to long term dependencies & related work**

Dear authors,

Thank you very much for your interesting contribution!

Some time ago I tried to contact you by mail but never got a reply. Therefore, I've decided to posted here. In particular, I wanted to point you to a few papers that I think are quite related, but are not discussed / cited:

1. CKConv ( https://arxiv.org/abs/2102.02611 ) introduces the SC raw dataset and is another approach able to model long term dependencies at every layer. Subsequently, both FlexNets ( https://arxiv.org/abs/2110.08059 ) and CCNNs (https://arxiv.org/abs/2206.03398) report results in this dataset as well (in fact CCNN still outperforms MEGA).

2. FlexConv introduces a masking mechanisms with a Gaussian mask. From what I understand this mechanism is very related to the EMA mechanism used in your paper. We observe that FlexConv brings important improvements over CKConv, alike to those observed in your paper for self-attention.

3. In addition, we would like to note that CCNNs are, to the best of our knowledge, competitive with S4, and thus, within the best existing approaches. Note that this is also a convolutional architecture and therefore its complexity behave much better than self-attention. Consequently, we believe that this architecture is very relevant to the comparisons performed in your paper e.g. in the LRA dataset. Moreover, and in contrast to S4, the CCNN is to the best of our knowledge the only long term network that can be directly applied to $\mathbb{R}^N$ data. Given that you also tackle tasks on images, we thing that a pointer there would be interesting and help the reader.

We would sincerely appreciate if you could discuss / cite these papers in case you find them relevant.

Best regards,

David

---

> ### Author Response · Authors · 2022-11-09
> **Re: Long term dependencies & related work**
>
> Hi David,
>
> Thanks for sharing with us these related works and sorry for missing your previous email.
> We have revised our paper to cite these papers and discuss them in the related work section.

---

### Comment · Reviewer_Dc4o · 2022-11-08
**Sorry for the mid-rebuttal update**


I updated my original review, because I realized recently that I may have some concerns with theorem 1.

---

> ### Author Response · Authors · 2022-11-09
> **Response to the concern of Theorem 1**
>
> We really appreciate your comments on Theorem1 and try to address your concern in the following thread:
> https://openreview.net/forum?id=qNLe3iq2El&noteId=nGud2_UurK

---

### Author Response · Authors · 2022-11-09
**General Response on Ablation Studies**

Thank you again for taking time to review our paper and providing constructive feedback! We’ll address some common concerns below, mainly ablation studies on various factors of Mega. We will add a more formal discussion in the appendix to include these ablation studies.

For the ablation studies, we conducted experiments on two LRA tasks – Text (IMDB) and Image (CIFAR-10).

1. Effect of damping factor in EMA
| | w/ damping factor in EMA | w/o damping factor in EMA |
| -------- | -------- |  -------- |
| Text    |  90.43   |   89.98       |
| Image   | 90.44   |   89.21  |

We find that removing the damping factor can consistently and slightly decrease the performance on both text and image datasets.

2. Effect of removing (relative) positional information
| | w/ relative positional bias| w/o relative positional bias |
| -------- | -------- |  -------- |
| Text    |  90.43   |   90.16   |
| Image   | 90.44   |   75.21  |

We find that removing relative positional bias can have significant negative effects on image data but only worsen the performance on text slightly. We hypothesize this is because the LRA image task flattens each image as a sequence of pixels by sacrificing spatial information, which makes the relative positional bias important to recover the spatial information.

3. Replacing EMA with other S4 variants: S4(D) + attention (EMA vs. S4D vs. S4)

|     |                    | EMA   | S4D   | S4     |
| --- | ------------------ | ----- | ----- | ------ |
| Text    | Mega               | 90.43 | 89.93 | 89.78  |
|     | Mega-chunk (c=128) | 90.19 | 89.12 | 89.23  |
| Image    | Mega               | 90.44 | 90.12 | 90.21  |
|     | Mega-chunk (c=128) | 85.80 | 85.06 | Failed |

We replaced EMA in Mega and Mega-chunk with S4D and S4 respectively and found that S4D and S4 didn’t bring any performance gains on the two LRA datasets. Additionally, training with S4 can lead to unstable training.

---

> ### Comment · Reviewer_Dc4o · 2022-11-09
> **S4 vs EMA question**
>
> Just to be clear, does both S4/S4D and EMA when tested within MEGA/MEGA-chunk use bidirectionality for fairness?

---

> > ### Author Response · Authors · 2022-11-09
> > **Response to S4 vs EMA question**
> >
> > Yes, we did. We also used bidirectional S4/S4D when performing the experiments.

---

> ### Public Comment · ~Albert_Gu1 · 2022-11-09
> **Details for the SSM comparisons**
>
> Thanks for running these ablations! Many researchers are very intrigued by the fact that EMA works so well despite several papers observing that real-valued SSMs perform worse than complex-valued SSMs (as you noted in other responses, this is the main difference between EMA and S4D). If you don't mind, could I check some details to ensure the best possible comparison:
>
> - Are you using the original version of S4, or the most recent version? The original version had potential instability problems that were fixed in work published in Feb 2022. The best version to use out-of-the-box is this [file](https://github.com/HazyResearch/state-spaces/blob/main/src/models/s4/s4.py) ([README](https://github.com/HazyResearch/state-spaces/tree/main/src/models/s4) for usage). You would pass `mode=diag measure=diag-lin` into the layer for S4D
> - The learning rate on recurrent parameters is supposed to be lowered, which is a more general phenomenon that helps RNNs. If EMA doesn't do this, it would be reasonable to not have it for S4, but this would disadvantage the baseline. The easiest compromise might be to **make the SSM parameters non-trainable** by passing in `lr={'A': 0.0, 'B': 0.0}` into the layer (you don't need a custom optimizer for this)
> - As Reviewer Dc4o noted, other hyperparameters like bidirectionality should be the same. Other flags should also map 1-1 between the models, for example the EMA heads $h$ is the same as the `d_state` option in S4(D). To double check that everything is controlled, can you confirm that **EMA and S4D should have exactly the same speed** since EMA is implemented the same way as S4D. Additionally, if using the CUDA extension then **S4 should be faster and more memory efficient than EMA**.
> - Finally, several researchers have independently observed that **a complex-valued SSM usually dominates a real-valued SSM throughout training**, but it's possible that the real-valued one catches up later if tuned well. I think people would be really interested in seeing if EMA is consistently as good as S4(D). For the most compelling comparison, could you **include training/validation curves throughout training for these ablations**? I understand that producing a full Python plot can be cumbersome, so even a quick screenshot of the curves uploaded to an image hosting site like imgur would be great.
>
> I hope this is reasonable since it's just using a drop-in replacement where the flags have 1-1 correspondences, and it would be very convincing to show these plots!

---

> > ### Author Response · Authors · 2022-11-09
> > **Response to the details of SSM comparisons**
> >
> > Hi Albert,
> >
> > Thanks for your comments!
> > Here are some details of our experiments to answer your questions:
> >
> > > Are you using the original version of S4, or the most recent version? The original version had potential instability problems that were fixed in work published in Feb 2022. The best version to use out-of-the-box is this file (README for usage). You would pass mode=diag measure=diag-lin into the layer for S4D
> >
> > We were using the most recent version of S4 and S4D. We were using the `measure=legs` for S4 and `measure=inv` for S4D.
> >
> > > The learning rate on recurrent parameters is supposed to be lowered, which is a more general phenomenon that helps RNNs. If EMA doesn't do this, it would be reasonable to not have it for S4, but this would disadvantage the baseline. The easiest compromise might be to make the SSM parameters non-trainable by passing in lr={'A': 0.0, 'B': 0.0} into the layer (you don't need a custom optimizer for this)
> >
> > We did not use different learning rate for S4(D) parameters, because EMA does not do it, either. We agree that it would be reasonable to do it for S4(D) to guarantee that these baselines are in their best mode. We will re-run these experiments by specifying smaller learning rate (e.g. $1e-3$) to SSM parameters.
> >
> > > As Reviewer Dc4o noted, other hyperparameters like bidirectionality should be the same. Other flags should also map 1-1 between the models, for example the EMA heads  is the same as the d_state option in S4(D). To double check that everything is controlled, can you confirm that EMA and S4D should have exactly the same speed since EMA is implemented the same way as S4D. Additionally, if using the CUDA extension then S4 should be faster and more memory efficient than EMA.
> >
> > Yes, we were using the same hyper-parameters, except $N$ of SSM. For EMA, as mentioned in the paper, we set $N=16$. For S4 and S4D, we set $N=32$, i.e. 16 for the real part and 16 for the imaginary part. And we confirmed that S4D has the same speed. We did not use the CUDA extension so S4 in our experiments is slightly slower than S4D and EMA.
> >
> > > Finally, several researchers have independently observed that a complex-valued SSM usually dominates a real-valued SSM throughout training, but it's possible that the real-valued one catches up later if tuned well. I think people would be really interested in seeing if EMA is consistently as good as S4(D). For the most compelling comparison, could you include training/validation curves throughout training for these ablations? I understand that producing a full Python plot can be cumbersome, so even a quick screenshot of the curves uploaded to an image hosting site like imgur would be great.
> >
> > We are also surprised that S4D did not obtain better results in our experiments when combined with attention. I guess one possible reason is that EMA restricts the parameters in $(0, 1)$, which explicitly controls the scale of its outputs. Attention-based models are usually sensitive to the scale of their hidden states. Another possible reason is that attention is doing the heavy-lifting parts which neutralizes the advantages of S4(D) over EMA. We are working on running a pure EMA model by entirely removing the attention part to compare with S4(D). I believe these results would help us to understand better. For the training/validation curve plots, we will update with a link in this thread soon.

---

> > > ### Public Comment · ~Albert_Gu1 · 2022-11-09
> > > **Thanks**
> > >
> > > Thanks for the quick response! It looks like everything checks out, and adding those couple of details (1. LR lowered or non-trainable for SSM parameters; 2. EMA without attention; 3. training curves) would make for a pretty comprehensive set of ablations.
> > >
> > > > I guess one possible reason is that EMA restricts the parameters in $(0, 1)$, which explicitly controls the scale of its outputs. Attention-based models are usually sensitive to the scale of their hidden states.
> > >
> > > The discretization step of SSMs serves the same role; it restricts $\bar{A}$ (the analog of the EMA $\alpha$) to have magnitude less than $1$. Actually, the sigmoid parameterization of $\alpha$ can be seen as a special case of discretization, which is a cool connection ([LSSL, Lemma 3.1](https://arxiv.org/abs/2110.13985)). So this is probably not the reason, but it's probably beyond the scope of this work to understand all the nuances - the ablations will leave a lot of interesting questions for future researchers.

---

> > > > ### Author Response · Authors · 2022-11-09
> > > > **Thanks for your feedback!**
> > > >
> > > > We are working on the these three experiments/plots you mentioned right now and will update these results in this thread soon. Hopefully, all of our ablation studies are interesting and helpful to all the readers.

---

> ### Comment · Reviewer_ZPfy · 2022-11-11
> **Question on Ablations**
>
> Thank you for running these ablations, they are useful to compare EMA with S4D and S4. I look forward to seeing the results from Albert's suggested experiments.
>
> One question: do you have results on the other tasks in LRA? I would be curious to see those as well to see if the trends above hold.

---

> > ### Author Response · Authors · 2022-11-11
> > **Response to Ablations**
> >
> > Thanks for asking! We will update this thread with these results and on other tasks in LRA over the weekend.

---

> ### Author Response · Authors · 2022-11-15
> **Additional Experimental Results**
>
> We have followed Albert's suggestions, we conducted the following experiments:
>
> 1. Set a lower LR (1e-3) for the parameters of SSM (`A` and `B`) with weight decay `0.0`.  As suggested by ***ZPfy***, we added experiments for PathFinder (`32x32`)
>
> Replacing EMA with other S4 variants: S4(D) + attention (EMA vs. S4D vs. S4)
>
> |     |                    | EMA   | S4D   | S4D (low LR) | S4 | S4 (low LR) |
> | --- | ------------------ | :-----: | :-----: | :------: | :------: | :------: |
> | Text    | Mega                   | 90.43 | 89.93 | 89.90 | 89.78  |  89.81  |
> |     | Mega-chunk (c=128) | 90.19 | 89.12 | 89.21  | 89.12 | 89.11  |
> | Image    | Mega               | 90.44 | 90.12 | 90.11  | 90.21 | 90.12 |
> |     | Mega-chunk (c=128) | 85.80 | 85.06 | 85.37 | Failed| 85.01 |
> | PathFinder   | Mega         | 96.01 | -- | 95.98  | -- | 95.92 |
> |     | Mega-chunk (c=128) | 94.41 | -- | **94.62** | --| 94.34 |
>
> We see that with lower LRs, S4D and S4 became more stable. But the final accuracy has not been improved significantly.
>
> On PathFinder, we observed slight improvement on Mega-chunk by replacing EMA with S4D.
>
> 2. Training curves
>
> We uploaded the training curves of Mega and Mega-chunk with EMA, S4D and S4 on CIFAR dataset.
> https://anonymous.4open.science/r/lra_plots-39EA/
>
> 3. EMA without attention (integrating EMA into the SSM framework to replace S4(D))
>
> We also ran experiments on the three LRA tasks: IMDB (Text), CIFAR-10 (Image) and PathFinder. For EMA, we exactly followed the hyper-parameters reported in Gu et al., 2022, except by increasing the warmup steps in CIFAR-10 and PathFinder.
> For S4 we used `mode=nplr` and `measure=legs`, and for S4D, we used `mode=diag` and `measure=diag-lin`.
>
> |         | EMA   | S4D   |  S4 |
> | ----- | :------: | :-----: | :------: |
> | Text             | 87.02 | 86.59 | 86.87 |
> | Image          | 80.82 | 86.62 | 88.36 |
> | PathFinder  | Failed | 94.13 | 92.75 |
>
> We see that EMA is able to achieve competitive accuracy on IMDB. On CIFAR-10, EMA is worse than S4D and S4. On PathFinder, EMA failed to converge. One possible reason is we did not properly tune the hyper-parameters of EMA.

---

> > ### Public Comment · ~Albert_Gu1 · 2022-11-17
> > **Comments on additional ablations**
> >
> > Thanks for providing these ablations! The results with the pure SSM blocks without attention align with what I and other researchers have found, that real SSMs (EMA) are noticeably worse than complex SSMs (S4/D). The results with SSM variants inside the full Mega block are surprising and interesting. Like you said, it's possible that attention plays a bigger role and the effect of the SSM is diminished with the full Mega model.
> >
> > One thing that stands out to me are the results on Mega-chunk. I would expect that as the chunk size decreases, EMA starts performing worse than S4(D), because in the limit $c=1$ this is just an attention-free model. It seems to me that chunk size $c=128$ should be somewhere in between these regimes, so the results showing parity are surprising.
> >
> > I'm quite curious about these findings, so I incorporated the provided code into the S4 codebase and will try to reproduce some of them. (Update: results provided in next post)

---

> > ### Public Comment · ~Albert_Gu1 · 2022-11-17
> > **Reproductions of Ablations**
> >
> > I have attempted to reproduce these ablations.
> > All code, experiment config files, and a README describing them in more detail has been released publically [here](https://github.com/HazyResearch/state-spaces/tree/mega/configs/experiment/mega/lra-image).
> >
> > - I incorporated the Mega block and EMA code into the state-spaces repo, which is a general framework for training sequence models.
> > - I focused on the LRA-Image dataset and did not experiment with any others. I focused on Mega-chunk ($c=128$) for the reasons described in my last reply.
> > - All parameters specified in the Mega paper and experiment script were recreatable in my codebase, so the models match as far as I can tell (except possibly in minor differences in the embedding and classification head layers, which shouldn't make a big difference).
> > - For faster ablations, I also experimented with a smaller model that halved the depth, width, training steps, and regularization (details in README)
> >
> > -------------
> >
> >   Here are my results on 3 block designs (large Mega block, small Mega block, small vanilla block with no gating or attention).
> >
> > | Model                 | Params | Val Acc |
> > |-----------------------|--------|---------|
> > | (large) Mega-EMA      | 2.65M  |   82.60 |
> > | (large) Mega-S4D-Real | 2.65M  |   83.98 |
> > | (large) Mega-S4D      | 2.65M  |   84.68 |
> > | | | |
> > | (small) Mega-EMA      | 279K   |   79.98 |
> > | (small) Mega-S4D-Real | 279K   |   81.62 |
> > | (small) Mega-S4D      | 279K   |   80.80 |
> > | | | |
> > | (small) EMA           | 267K   |   70.74 |
> > | (small) S4D-Real      | 200K   |   70.34 |
> > | (small) S4D$^*$        | 200K   |   84.40 |
> >
> >  $^{*}$ differences in parameter count are due to a small implementation detail involving weight-tying, which does not affect speed or performance. These have been corrected in later ablations with an exact parameter-matched model.
> >
> > **[Training and validation curves](https://github.com/HazyResearch/state-spaces/blob/mega/configs/experiment/mega/lra-image/mega_ablations_1000_warmup_all.pdf)**
> >
> >
> > ------------
> >
> > **TL;DR reproduction results:**
> > - My Mega reproductions in the state-spaces repo match all details of the model description as far as I can tell, but have a small gap in performance (82.5 vs 86) to the reported results.
> > - My results confirm that EMA is much worse than (complex-valued) S4D when they are used interchangeably in a generic DNN.
> > - My results show that with Mega-*chunk* ($c=128$), EMA is still consistently worse than S4D across different model sizes. I think this is intuitive given the huge gap for a vanilla model without attention ($c=1$), but contradicts the findings in your curves.
> > - I have released my reproduction code and experiments so that others may confirm or point out errors in my reproduction.

---

> > > ### Author Response · Authors · 2022-11-17
> > > **Re: Reproductions of Ablations**
> > >
> > > Hi Albert,
> > >
> > > We really appreciate your results! According to your results, (large)-Mega with EMA and S4D are below what we have obtained (82.6/84.7 vs. 86). Thus, we guess there are some hyper-parameters that were not properly set and Mega (both with EMA and S4D) was not running in its optimal mode. I have read your training scripts and found that might be the `warmup steps`. In our experiments, we set `warmup=9000` for CIFAR-10 in LRA.
> > >
> > > In fact, when we integrated EMA into the SSM codebases, we also increased the `warmup steps` for EMA on CIFAR-10. With your default setting `warmup=1000`, EMA without attention can only obtain 67% accuracy. By increasing `warmup=9000`, the same model obtained 80.8% accuracy. We guess if we keep tuning other hyper-parameters, like `learning rate` and `weight decay`, we can narrow the gap between EMA and S4(D).

---

> > > > ### Public Comment · ~Albert_Gu1 · 2022-11-17
> > > > **Thanks**
> > > >
> > > > Thanks for pointing out the discrepancy so quickly. I'll run it again and dig in more and post an update when results come back. I'll also try to run S4D from your LRA-Image script training pipeline.
> > > >
> > > > To be clear, the settings for both EMA and S4D are exactly equal in all ways for these ablations. So at the minimum this suggests that EMA is much harder to tune, all else being equal. `warmup=1000` is likely not optimal for S4D either, it was just never tuned.

---

> > > > > ### Author Response · Authors · 2022-11-18
> > > > > **Response**
> > > > >
> > > > > > To be clear, the settings for both EMA and S4D are exactly equal in all ways for these ablations. So at the minimum this suggests that EMA is much harder to tune, all else being equal. warmup=1000 is likely not optimal for S4D either, it was just never tuned.
> > > > >
> > > > > In our ablation study of Mega with EMA and S4(D), we also have exactly equal setting (`warmup=9000` for all the model variants), and Mega with EMA obtained similar accuracy of Mega with S4(D). For the training of attention-based model, it is very common to properly set `warmup steps` to be long enough to ensure stable convergence of training during the beginning stage. With that being said, we cannot conclude that EMA is harder to tune, as other hyper-parameters, such as `learning rate`, `weight decay` etc.,  were tuned towards S4(D). And EMA just uses random initialization, which is also simpler than S4(D) which relies on HiPPO.

---

> > > > > > ### Public Comment · ~Albert_Gu1 · 2022-11-18
> > > > > > **Not correct**
> > > > > >
> > > > > > > In our ablation study of Mega with EMA and S4(D), we also have exactly equal setting (warmup=9000 for all the model variants), and Mega with EMA obtained similar accuracy of Mega with S4(D)
> > > > > >
> > > > > > Actually, I'm wondering if the model was implemented correctly; there is a subtlety to pay attention to, which is that the S4 convolution block has optional additional components such as a linear mixing layer and also uses different activation functions than Mega. To make it exactly interchangeable with Mega, some small adjustments are needed. See the following lines  in my implementation: [1](https://github.com/HazyResearch/state-spaces/blob/bc86032a5dc7c51c68037196c9a5337f15a559f9/src/models/sequence/ss/ema.py#L220) [2](https://github.com/HazyResearch/state-spaces/blob/bc86032a5dc7c51c68037196c9a5337f15a559f9/src/models/sequence/mega.py#L98) (the `linear=True` flag) and [3](https://github.com/HazyResearch/state-spaces/blob/bc86032a5dc7c51c68037196c9a5337f15a559f9/src/models/sequence/mega.py#L262)
> > > > > >
> > > > > > > With that being said, we cannot conclude that EMA is harder to tune, as other hyper-parameters, such as learning rate, weight decay etc., were tuned towards S4(D).
> > > > > >
> > > > > > It's a common misconception that S4(D) is highly tuned. In fact, replacing a few of the flags with the ones from Mega (e.g. warmup steps, and normalization) with *no extra tuning* has improved the best S4D results (from 86 to 88 on LRA-Image). I will post these tomorrow. In all settings I have been testing, S4D has been better than EMA when the architectures and hyperparameters were carefully controlled.
> > > > > >
> > > > > > > And EMA just uses random initialization, which is also simpler than S4(D) which relies on HiPPO.
> > > > > >
> > > > > > **S4D does not require HiPPO**, which is another common misconception. The initialization is also extremely simple, where the eigenvalues are initialized linearly.
> > > > > >
> > > > > > Even if the initialization is more complicated, I don't think that changing the initialization is enough to constitute a "new method".
> > > > > >
> > > > > > Finally, S4D actually already introduced a real-valued variant called S4D-Real which has been published. Thus **Mega's EMA is not the first multi-head EMA method**. The initialization is extremely simple, which simply initializes $A = \text{diag}(-1, -2, -3, ..., -N)$. In my ablations, this variant is also consistently outperforming EMA while being worse than the complex S4D. These will be posted when runs finish overnight.

---

> > > > > > > ### Author Response · Authors · 2022-11-18
> > > > > > > **Re: Not correct**
> > > > > > >
> > > > > > > >Actually, I'm wondering if the model was implemented correctly; there is a subtle to pay attention to, which is that the S4 convolution block has optional additional components such as a linear mixing layer. To make it exactly interchangeable with Mega, you have to turn off this layer.
> > > > > > >
> > > > > > > When implementing S4(D) in Mega, we just replaced the kernel of EMA with the kernel of S4(D), leaving other components unchanged. We believed that this is the correct way to compare Mega with EMA and S4(D). After reading your implementation and the optional additional components you referred to, we confirmed that our implementation is correct. In your reported results, Mega-chunk with EMA is much worse than that reported in our paper (82 vs 86). We are looking forward to your new results.
> > > > > > >
> > > > > > > > It's a common misconception that S4(D) is highly tuned. In fact, replacing a few of the flags with the ones from Mega (e.g. warmup steps, and normalization) with no extra tuning has improved the best S4D results (from 86 to 88 on LRA-Image). I will post these tomorrow. In all settings I have been testing, S4D has been better than EMA when the architectures and hyperparameters were carefully controlled.
> > > > > > >
> > > > > > > We are not saying that S4(D) is highly tuned. What we want to stress is that **Mega is not more heavily tuned than S4(D)**. As we discussed in previous threads, most of the hyper-parameters were directly borrowed from S4-v1 without tuning for fair comparison. It is possible that Mega can obtain even better performance, if we also tune these hyper-parameters, such as number of layers, model dimensions etc.
> > > > > > >
> > > > > > > > S4D does not require HiPPO, which is another common misconception. The initialization is also extremely simple, where the eigenvalues are initialized linearly.
> > > > > > >
> > > > > > > Thanks for pointing this out.
> > > > > > >
> > > > > > > > Finally, S4D actually already introduced a real-valued variant called S4D-Real which has been published. Thus Mega's EMA is not the first multi-head EMA method. The initialization is extremely simple, which simply initializes . In my ablations, this variant is also consistently outperforming EMA while being worse than the complex S4D.
> > > > > > >
> > > > > > > As we claimed in our paper and replied in our previous response, the main motivation of Mega is to incorporate stronger inductive bias by the classic EMA approach, **not** a method for combining state space models with attention. Neither finding the optimal SSM formulation in Mega nor demonstrating EMA without attention is better than S4(D) is central to our paper. Thus, we believe this ablation study is, though helpful for us to better understand Mega, not central to our paper.
> > > > > > >
> > > > > > > At last, S4D is a concurrent work of Mega. We have completed the design and implementation of Mega before S4D was released. In addition, we want to say that **EMA can be but not have to be regarded as an instantialization of SSM**. In its own formulation, EMA is more straight-forward to be viewed as an average smoothing approach for sequential data.

---

### Public Comment · ~Albert_Gu1 · 2022-11-19
**Mega, EMA, S4D Ablations**

As discussed by some reviewers and in other threads in this forum, Mega’s core EMA layer is closely related to prior methods such as S4D (more details provided in reply to this post). During the rebuttal phase, the authors ran ablations on these methods. I have independently run some of these ablations, as well as reproduced the Mega model in another codebase and performed many more ablations.

## Ablations in Mega codebase
Since the earlier thread, I have run additional ablations in the official Mega codebase using their train script, swapping out the inner EMA layer with interchangeable baselines such as S4D. The dataset is LRA-Image (grayscale CIFAR-10) with chunk size $c=128$. The baselines were run out-of-the-box with no modifications or tuning, and *all hyper-parameters match those for the best-performing Mega-EMA model*.

In addition to the default S4D-Lin model, I have included another variant called **S4D-Real** which was published as an ablation to S4D. This masks the model so that the $A$ matrix is always real-valued, which is a *prior "multi-head EMA" almost identical to Mega's EMA layer with only minor differences in the parameterization*.


| Model                | Params   |   s/epoch |   Val Acc |
| -------------------- | -------- | --------- | --------- |
| Mega-EMA (original)  | 2.82M    |       195 |     86.10 |
| Mega-S4D-Real        | 2.74M    |       152 |     87.00 |
| Mega-S4D             | 2.74M    |       152 |     87.12 |

**[Training and validation curves](https://github.com/HazyResearch/state-spaces/blob/e9ce652126cc773dcb6bb7d6f7270c425d4a36a2/configs/experiment/mega/lra-image/mega_ablations_mega_repo.pdf)**

## Ablations in S4 codebase

I have reproduced the Mega model and run several ablations on various model sizes and architectures.
All code, experiment config files, and a README describing them in more detail has been released publically **[here](https://github.com/HazyResearch/state-spaces/tree/ede0b53fe4bcfccf185c32b99880463b2a2cd085/configs/experiment/mega/lra-image)**.

Outside of the inner EMA/S4D layer, *all hyperparameters are matched to the best-performing hyperparameters of the Mega-EMA model* specified by the paper. All results are 1 single run with *no tuning* for the baselines.



### Large Mega model

The following models attempt to reproduce the original Mega model above. Details in reply.

| Model                   | Params   | s/epoch   | Val Acc   |
| --------------------    | -------- | --------- | --------- |
| (large) Mega-EMA        | 2.73M    | 180       | 82.56     |
| (large) Mega-EMA-Repro^ | 2.65M    | 124       | 83.42     |
| (large) Mega-S4D-Real   | 2.65M    | 121       | *84.44*   |
| (large) Mega-S4D        | 2.65M    | 122       | **86.22** |

^ This is a slightly refactored version of the original Mega module, in a way that parameter matches the S4D baselines and allows swapping them in easily, that in fact seems to have improved and sped up the model. Details in reply.


### Small Mega model

The model depth, width, training/warmup steps, and weight decay were halved. A single run was performed with no other experimentation or tuning.

| Model                  | Params   | s/epoch   | Val Acc   |
| --------------------   | -------- | --------- | --------- |
| (small) Mega-EMA       | 299K     | 51        | 81.16     |
| (small) Mega-EMA-Repro | 279K     | 51        | 80.76     |
| (small) Mega-S4D-Real  | 279K     | 54        | *81.20*   |
| (small) Mega-S4D       | 279K     | 53        | **81.46** |

### Large vanilla model (no gating or attention)

The following models use a basic convolution block to investigate the interchangeable inner layer (EMA vs S4D) in isolation.

| Model                | Params   |   s/epoch | Val Acc   |
| -------------------- | -------- | --------- | --------- |
| (large) EMA          | 4.35M    |       128 | 70.96     |
| (large) EMA-Repro    | 3.96M    |       119 | 71.52     |
| (large) S4D-Real     | 3.96M    |       105 | *74.30*     |
| (large) S4D          | 3.96M    |       105 | **88.28**   |



### Small vanilla model (no gating or attention)

| Model                  | Params   | s/epoch   | Val Acc   |
| --------------------   | -------- | --------- | --------- |
| (small) EMA            | 333K     | 31        | 69.96     |
| (small) EMA-Repro      | 267K     | 30        | 69.38     |
| (small) S4D-Real       | 267K     | 32        | *70.88*   |
| (small) S4D            | 267K     | 30        | **82.78** |

**[All training and validation curves](https://github.com/HazyResearch/state-spaces/blob/e9ce652126cc773dcb6bb7d6f7270c425d4a36a2/configs/experiment/mega/lra-image/mega_ablations_10000_warmup_all.pdf)**

---

> ### Public Comment · ~Albert_Gu1 · 2022-11-19
> **Comments on Mega Paper**
>
> Since I have now worked closely with the Mega code, performed many experimental ablations, and read the submission carefully, I'll share a more direct opinion about the framing of this paper.
>
> After reading the Mega code carefully, it is clear to me that **Mega's EMA is heavily based on the code of S4 and S4D**. Although EMA and SSMs (state space models) are more general concepts, S4(D) has many specific design choices and implementation details that Mega’s MultiHeadEMA borrows. I will support this claim with details in the next reply.
>
> There is of course nothing wrong with being a close variant of prior methods, and I'm very glad that the S4 line of work has inspired ongoing research. But this does raise serious questions about the framing of the paper that some reviewers have also brought up.
>
> 1. Sharing Reviewer ZPfy's concern, I don't think that the framing of Mega as inspired by the "classical EMA" is a fair depiction of the actual lineage of ideas it is based on.
>
> 2. From an empirical point of view, these new ablations seem to suggest that **Mega's EMA is strictly worse than its predecessors**, or at the very least much harder to tune.
>
> 3. One could try to make the argument that the EMA is simpler, but there are also no differences in efficiency or code complexity ([EMA](https://github.com/HazyResearch/state-spaces/blob/e9ce652126cc773dcb6bb7d6f7270c425d4a36a2/src/models/sequence/ss/ema.py#L15)  vs. [S4D](https://github.com/HazyResearch/state-spaces/blob/e9ce652126cc773dcb6bb7d6f7270c425d4a36a2/src/models/s4/s4d.py#L68))
>
> 4. Additionally, **Mega is not even the first multi-head EMA**, which is one of the main claimed contributions of the paper. Both S4D and its predecessor [DSS](https://arxiv.org/abs/2203.14343) ablated real-SSM variants which are *exactly a "multi-head EMA" almost identical to Mega's*. As the previous ablations showed, these variants (S4D-Real) are also consistently better than Mega's version of MultiHeadEMA.
>
> 5. To me, an important bottom line question of any paper is: **When and why would an ML practitioner want to use the proposed method (e.g. multihead EMA) over alternatives?** I don't think this has been addressed in the paper because of the framing: I suspect that most readers would not even have a clear picture of what the drop-in alternatives are. And based on the extensive ablations that have been performed, thus far there does not seem to be a good reason to use this formulation of EMA over the prior work that it is based on, including an *already published and better version of multi-head EMA*.
>
> To be clear, aside from the "multi-head EMA" part, in my opinion this would be a fine submission that proposes several architectural ideas with extensive empirical contributions. However, given the above facts and how prominently EMA is part of the framing of the paper, I'm not sure that it is positioned in a way that is most constructive for the broader community - for example, to an applied ML researcher who just wants to understand how methods in this area relate to each other and implement the best model for their problem. The most confusing part of all this is that it seems like the proposed Mega method could have had substantially better results with less work, by just using the original methods that it is based on.
>
> Of course, these are just my opinions and perhaps the reviewers and other readers will have different perspectives.

---

> > ### Public Comment · ~Albert_Gu1 · 2022-11-19
> > **Mega vs S4(D) code examples**
> >
> > This post provides examples of details to support the claim that Mega is largely based on the code of S4(D). Because I cannot link to specific lines in the supplementary code, or to lines in the official Mega repository without breaking the authors' anonymity, I will link to my own reproductions of their modules. These are faithful transcriptions of the original Mega code with the same code execution path, with only some variable names changed back to their originals.
> >
> >
> > ### Computing the EMA recurrence with FFT-convolution
> > [[Mega](https://github.com/HazyResearch/state-spaces/blob/e9ce652126cc773dcb6bb7d6f7270c425d4a36a2/src/models/sequence/ss/ema.py#L152)] [[S4](https://github.com/HazyResearch/state-spaces/blob/e9ce652126cc773dcb6bb7d6f7270c425d4a36a2/src/models/s4/s4d.py#L104)]
> >
> > Viewing the EMA recurrence $x \gets \alpha x + \beta u$ as a convolution does not appear in the EMA modeling literature. This particular way of **computing recurrences as convolutions** is introduced by S4, and the general FFT-convolution approach has appeared in a broader line of work on wide-kernel convolutions mentioned earlier in this forum by David Romero.
> >
> > ### Bidirectional convolution kernel
> > [[Mega](https://github.com/HazyResearch/state-spaces/blob/e9ce652126cc773dcb6bb7d6f7270c425d4a36a2/src/models/sequence/ss/ema.py#L143)] [[S4](https://github.com/HazyResearch/state-spaces/blob/e9ce652126cc773dcb6bb7d6f7270c425d4a36a2/src/models/sequence/ss/s4.py#L190)]
> >
> > Making the model bi-directional in an efficient way is not obvious. Rather than computing a layer in the forward and reverse direction separately (requiring *two* calls to the module), S4 creates two copies of convolution kernels and stacks them back-to-back so that it can be computed in *one* FFT-convolution operation. **This technique is actually not published, yet Mega borrows the same implementation, which is somewhat non-trivial.**
> >
> > In fact, the official Mega code has a mistake in the implementation and **Mega-EMA is 30% slower end-to-end than Mega-S4D** in their official codebase, and 50% slower in mine. The authors should probably check that they copied the reference implementation correctly (in particular pay attention to the tensor shapes and lengths). Benchmark times are provided in the ablation post.
> >
> > ### Kernel caching logic
> > [[Mega](https://github.com/HazyResearch/state-spaces/blob/e9ce652126cc773dcb6bb7d6f7270c425d4a36a2/src/models/sequence/ss/ema.py#L102)]
> > [[S4](https://github.com/HazyResearch/state-spaces/blob/cfb6a06c3b3b61d97096d0154d5b6208e54a0f2d/src/models/sequence/ss/kernel.py#L651)]
> >
> > These global-convolution models generally consist of two steps: (1) compute a convolution kernel from parameters (2) apply the convolution. A minor optimization is that step (1) can be cached during validation since the kernel doesn't change, a technique that Mega borrowed from the first version of S4.
> >
> > ### S4D: Vandermonde kernel
> > [[Mega](https://github.com/HazyResearch/state-spaces/blob/e9ce652126cc773dcb6bb7d6f7270c425d4a36a2/src/models/sequence/ss/ema.py#L88)]  [[S4D](https://github.com/HazyResearch/state-spaces/blob/e9ce652126cc773dcb6bb7d6f7270c425d4a36a2/src/models/s4/s4d.py#L49)]
> >
> > A dead giveaway that Mega is based on S4D is the implementation of the Vandermonde kernel, the computational core of S4D. The term "Vandermonde" in the context of SSMs - and the implementation with this particular `arange-exp-einsum` pattern and these particular tensor shapes - has only appeared in S4D. Notably, despite using the term "Vandermonde product" in the paper and borrowing the core implementation, the original Mega submission does not even mention S4D.
> >
> > Claiming that S4D is concurrent work, as the authors have done elsewhere in this forum and in the revised paper, is misleading since **S4D is already published and the authors are explicitly using ideas and code from the paper**.
> >
> > ### Overall state space model (SSM) formulation
> > [[Mega](https://github.com/HazyResearch/state-spaces/blob/e9ce652126cc773dcb6bb7d6f7270c425d4a36a2/src/models/sequence/ss/ema.py#L50)]
> >
> > The parameters in Mega's EMA layer are explicitly modeled off of SSMs, with variables renamed from $(\Delta, A, B, C, D)$ in S4 terminology to $(\delta, \alpha, \beta, \gamma, \omega)$ in the Mega paper and code.
> >
> > As previously discussed, Mega's EMA layer is implemented exactly like S4(D), as a Vandermonde kernel constructed from $\Delta, A, B, C$ (i.e. $\delta, \alpha, \beta, \gamma$).
> >
> > [[Mega](https://github.com/HazyResearch/state-spaces/blob/e9ce652126cc773dcb6bb7d6f7270c425d4a36a2/src/models/sequence/ss/ema.py#L155)] [[S4](https://github.com/HazyResearch/state-spaces/blob/e9ce652126cc773dcb6bb7d6f7270c425d4a36a2/src/models/s4/s4d.py#L109)]
> >
> > Additionally, the convolution is followed by a skip connection multiplied by $D$ ($\omega$ in the Mega code). This skip connection detail is clearly derived from SSM models ([below eq. (1)](https://arxiv.org/pdf/2111.00396.pdf)) and is not even described in the Mega paper.

---

> > > ### Public Comment · ~Albert_Gu1 · 2022-11-19
> > > **Summary**
> > >
> > > [[Mega](https://github.com/HazyResearch/state-spaces/blob/e9ce652126cc773dcb6bb7d6f7270c425d4a36a2/src/models/sequence/ss/ema.py#L15)]  [[S4D](https://github.com/HazyResearch/state-spaces/blob/e9ce652126cc773dcb6bb7d6f7270c425d4a36a2/src/models/s4/s4d.py#L68)]
> > >
> > > **Overall, the code execution path of Mega's MultiHeadEMA layer and the S4D layer are nearly identical except for minor differences in parameterization.** Nearly every piece of the MultiHeadEMA code corresponds 1-1 to some version of S4(D).
> > >
> > > I again want to emphasize that there is nothing wrong with taking ideas from prior work, and I'm glad that others are building off this research. However this has not been appropriately positioned in this paper. The intention of these examples is to support the claim in the previous post that **Mega's EMA is not most closely related to the "classical EMA", but is a rebranding of recent work on deep SSMs**.
> > >
> > > I also want to emphasize that I believe that Mega has provided several other legitimate contributions. And even with the most pessimistic interpretation - that Mega's EMA is a strictly worse version of prior models such as S4(D) - I think it could still have been a reasonable component if framed well, e.g. as an ablation study on the SSMs it was derived from. However, the current version of the paper is (in my opinion) not very constructive - in fact even counter-productive - to the broader community, which is why I'm sharing this opinion publically.

---

> > > > ### Comment · Reviewer_Dc4o · 2022-11-19
> > > > **Some points**
> > > >
> > > > Thank you for your effort in bringing all these into light. I will reflect on this matter more but there are a few points I want to make clear on where I stand/and also a few questions so that I can make a better decision:
> > > >
> > > >
> > > > Some points related to my current stance and past decisions:
> > > >
> > > >
> > > > * I was roughly aware from the get go that the paper (despite that it makes the appearance of derivation from EMA) it takes heavy inspiration from S4 given the dimension-wise processing and reliance on FFTs. I was not too concerned about it initially because the authors are explicit that their EMA is a resticted version of S4 (although perhaps, the inspiration is underemphasized). And it is still interesting to see a simple variant with an EMA-related interpretation combined with GAU can perform as well.
> > > >
> > > >
> > > > * More concerning part is, as you said, is that the experimental trade-offs are a bit unclear (which was something I asked for in terms of ablations in my original review) - for example, what happens when EMA is replaced by S4 in MEGA, and the SSM block is replaced by EMA in S4 architecture?  The author seemed to adress some of these questions but I am again unclear about them if there were some mistakes (note I am talking about S4 at this point not S4D). Still, I believe EMA would be more efficient than the original S4?
> > > >
> > > >
> > > > * The connection between EMA and S4D is more tricky. I acknowledge, that in technical terms EMA is too close to being a variant of S4D to be framed as its own thing with only a few lines tucked later on to mention the connection (and even some of the justifications made in this discussion separating them seems to be wrong as well - regarding S4D non-reliance on HiPPo or relying on complex numbers). But the tricky part comes from the fact that S4D was posted in arXiv in 23 June 2022 and the ICLR guideline says:
> > > >
> > > > https://iclr.cc/Conferences/2023/ReviewerGuide
> > > >
> > > > >A: We consider papers contemporaneous if they are published (available in online proceedings) within the last four months. That means, since our full paper deadline is September 28, if a paper was published (i.e., at a peer-reviewed venue) on or after **May 28, 2022**, authors are not required to compare their own work to that paper. Authors are encouraged to cite and discuss all relevant papers, but they may be excused for not knowing about papers not published in peer-reviewed conference proceedings or journals, which includes papers exclusively available on arXiv. Reviewers are encouraged to use their own good judgement and, if in doubt, discuss with their area chair.
> > > >
> > > > Moreover, the matter becomes more tricky if the paper actually uses the github code for S4D, and implementation ideas from the paper (instead of independently coming up with them) and then posing EMA as a new thing even if under careful experiments it is worse and slower than S4D.
> > > >
> > > > In case, the aurhors are directly using S4D code and ideas to create an inferior version of it then I am not entirely sure how to factor that into my review given the guidelines. It's probably something on which I would wait for what AC/SAC have to say.
> > > >
> > > >
> > > > Related questions:
> > > >
> > > > 1. If we try to give the authors maximal benefit of the doubt (note that authors also explicitly saysthat they started working on EMA before S4D), how plausible it is for someone to start thinking of possible simplifications of S4, come up with the EMA motivation, and then independently come up with the idea of using Vandermond product (already starting from the ideas in S4) without noticing S4D?
> > > >
> > > > 2. Regarding the code. It would not be entirely surprising to me if they are building upon S4 codebase (although authors may need to assign credits better) given the similarity of S4 and EMA but directly using S4D code and posing it as a concurrent work definitely seems to be a questionable move (if that is what is being done). However, looking at the differences in S4D and EMA implementation of Vandermonde kernel, it's not entirely obvious to me (maybe other reviewers can check and comment too) that EMA is strictly copied off from the other. Wouldn't it be possible that the authors just independently came up with the idea of using Vandermonde product (use of Vandermonde is described in MEGA paper too - they are not hiding it) and built that within their copy of S4 framework? Some similarities would be expected given that ultimately both are implementating the same functional form and within an otherwise similar framework but I am not entirely sure if I should conclude from that that EMA just copies S4D code (and the ideas from the paper). Can you comment a bit on this concern and the plausibility of this alternate hypothesis?
> > > >
> > > > 3. Also can you comment on the trade-offs involved in S4 (not S4D) and EMA? I am assuming S4 eg. S4-LegS would be actually better in task performance (contrary to author's experiments given your better results with S4D) but perhaps, slower?

---

> ### Public Comment · ~Albert_Gu1 · 2022-11-19
> **Ablation Details**
>
> ## Mega codebase
> I swapped in the S4D kernel with no additional changes. I am happy to provide a fork of my changes to the Mega repo for proof after this discussion.
>
> ## S4 codebase
>
>
>
>
> **Reproduction of original Mega model**
>
> Although the results don't quite match the results from the Mega codebase (82.5 vs 86), the hyperparameters were matched as best they could with no additional tuning. Small differences in the implementations do exist:
> - The encoder/decoder (embedding/projection) layers between the inputs/outputs and model may be slightly different (which also explains an 8K parameter difference between the versions). I used sensible defaults from the S4 codebase (linear embedding, linear projection, mean pooling classification head) and did not experiment with these.
> - The learning rate scheduler uses a cosine decay, 10K warmup steps, 200K total steps. The Mega trainer uses linear decay, 9K warmup steps, 180K total steps.
>
> Since these discrepancies should be minor, the rest of the models are carefully controlled, and the rest of the hyperparameters match the Mega specifications as closely as possible, these results are still a fair comparison where Mega-EMA is substantially worse than Mega-S4D.
>
> **Model Notes**
>
> - Mega-EMA uses the original monolithic Mega module, which constructs an EMA kernel and applies an FFT convolution.
> - Mega-EMA-Repro factors it into a modular Convolution Block + EMA Kernel, and adds some options to parameter-match the model with S4D. This did not affect (or perhaps improved) performance, ensuring that the other ablations are fair.
> - Mega-S4D is the Convolution Block + S4D Kernel. Swapping out the interchangeable EMA/S4D kernels differs by just a few lines of code.
> - Mega-S4D-Real uses the published S4D-Real model and forces the $A$ matrix to be real, which is exactly a multi-head EMA model.
>
> **Ablations with different hyperparameters**
>
> Versions of these ablations were performed in an earlier thread, and the authors pointed out a difference in the `warmup_steps` hyperparameter. This has been changed, increasing from 1000 (the default from the S4 repository, which was never tuned) to 10000. In all settings, the models always receive the same hyperparameters.
>
>  **[All training and validation curves](https://github.com/HazyResearch/state-spaces/blob/e9ce652126cc773dcb6bb7d6f7270c425d4a36a2/configs/experiment/mega/lra-image/mega_ablations_1000_warmup_all.pdf)** for `warmup=1000` models.

---

> ### Author Response · Authors · 2022-11-19
> **Re: Mega, EMA, S4D Ablations**
>
> We appreciated your detailed ablation study, and are happy to see that Mega-chunk + S4(D) obtains (slightly) better accuracy (86.1 to 87.1) on LRA-Image. Since S4(D) is theoretically more expressive than EMA, it is not surprising to see that **with small chunk size $c=128$** replacing EMA with S4(D) improves Mega-chunk accuracy. However, Mega-chunk + S4(D) is still significantly worse than Mega + EMA on LRA-Image. In addition, on some other tasks in LRA and other benchmarks, the gap between Mega and Mega-chunk is less significant than that on LRA-Image.

---

> ### Author Response · Authors · 2022-11-20
> **Response**
>
> We replied to you in this thread: https://openreview.net/forum?id=qNLe3iq2El&noteId=hrB2Mww_dM
>
> Thanks!

---

### Author Response · Authors · 2022-11-19
**Response to Mega, EMA, S4D Ablations**

In the following comments: https://openreview.net/forum?id=qNLe3iq2El&noteId=8x_IEjJXpon, Albert together with Reviewer *ZPfy* are trying to stress that the EMA component of Mega is directly borrowed from S4D, both the idea and the code.

We decided to open a new thread to post our response as we believe that their comments need to be rectified and that our response is useful and necessary to the broader community.

First, we would like to declare unequivocally that **the design, development and implementation of the EMA component in our Mega architecture was finalized much earlier before the S4D paper was publicly released**. We can share our original dev repo link with the full timestamped commit history of the EMA component to prove this after the anonymity period. Therefore, it is not misleading to claim S4D as a concurrent work.

In fact, we have completed most of our LRA experiments even before the second version of S4  (S4-v2) was released. That is why we compared with the results from both versions in the Table 2 of our paper, and clearly discussed this in Section 4.1. Obviously, **it would have been impossible for us to complete the design of other important components in Mega, the execution of all our experiments and paper writing by the time of the ICLR deadline if Mega was based on S4D**.

We will provide detailed evidences to support our claim in the next reply.

Most importantly, we want to stress again that the main motivation and contribution of Mega is **not** to combine attention with different SSM variants. When we first set out to do this work, we were motivated by the framework of EMA, which far precedes S4 (or S4D). We agree that EMA can be viewed as a specific form of S4, but this was not our view when we first wrote this paper, and because of this
1. we don’t believe it is dishonest to keep the narrative in the current form.
2.  we honestly think it is a bit unfair that we are being asked to completely change the narrative of the paper to fit an alternative interpretation of the proposed method .

If anything, it seems that Albert’s interpretation of our framework would be a great *follow-up* paper to the current one, as it is clearly reinterpreting and expanding on the results we originally presented.

Independent of the connections to S4, it is straight-forward to view EMA as an average smoothing approach for sequential data. We leverage EMA’s view as a smoothing approach to incorporate stronger inductive bias into attention, and this is our main motivation. Based on this original motivation for Mega, **neither finding the optimal SSM formulation within the Mega framework nor demonstrating that EMA without attention is better than S4(D) is central to the claims made in our paper**. Thus, we believe Albert’s ablation study on directly comparing EMA with S4(D) without attention, though helpful for us to better understand Mega, is not central to our paper, which, as mentioned above, we wrote with very different motivations.

Therefore, in our opinion, your question *When and why would an ML practitioner want to use the proposed method (e.g. multihead EMA) over alternatives?* should be investigated in future work and we would be glad to see this happen. In addition, though Mega’s EMA **as an individual  component** is theoretically less expressive than S4(D), **we cannot directly draw the conclusion that Mega + S4(D) is strictly better than Mega + EMA**. From our experiments, Mega + S4(D) did not obtain significant improvements over Mega + EMA. A possible and reasonable explanation is that the attention mechanism is sufficiently strong to do the heavy-lifting jobs, neutralizing the benefits of S4(D) over EMA. We would also be glad that some future work would draw more convincing conclusions by conducting extensive and comprehensive experiments, rather than results on one single dataset from the LRA benchmark.

---

> ### Author Response · Authors · 2022-11-19
> **Response to Albert's code examples**
>
> ### Computing the EMA recurrence with FFT-convolution
> **We never claimed the FF-convolution of EMA is our contribution and that is why we put this part in Appendix A as a supplementary content**. As Albert stated in his comments, the general FFT-convolution approach has appeared in a broader line of work on wide-kernel convolutions. It is not hard for us to apply FFT-convolution for EMA.
>
> ### Bidirectional convolution kernel
> Albert claimed that **creating two copies of convolution kernels** for the bidirectional EMA is directly copied from S4(D)'s code, just because this technique is somehow non-trivial. He even claimed that we made a mistake during this copy of code which leads to some slow down of our implementation.
>
> First, we need to point out that **there are no mistakes nor bugs in our implementation**. In fact, last week when we were running the ablation study of Mega + S4(D), we also noticed this speedup in the bidirectional part. Then, we went through S4(D)'s implementation and finally found the reason, which is in fact a minor issue in Albert's implementation. We will briefly describe the issue with the following example:
>
> Suppose we have input $x = {x_1, x_2, \ldots, x_T}$, and the two set of parameters for the forward and backward EMA (or SSM variants) are $\alpha_F$ and $\alpha_B$ (we omit the other parameters for simplicity). For bidirectional EMA, our implementation exactly computes the *correct* output $y_t = y_t^{F} + y_t^B$  where
>
> $y_t^F = \alpha_F \odot x_t + (1 - \alpha_F) \odot y_{t-1}^F$ and $y_t^B = \alpha_B \odot x_t + (1 - \alpha_B) \odot y_{t+1}^B$
>
> Albert's implementation is actually computing a slightly different output $y_t = y_t^F + y_{t+1}^B$, by shifting the backward pass one position. His implementation reduces the length of FFT computation, which leads to the speed up.
>
> **The distinction between the two implementations is not our mistake or bug, but direct evidence to prove that our bidirectional EMA was implemented independently.** We would like to re-emphasize that there is not any truth to these allegations that the code was copied (and can provide evidence to demonstrate this). While plagiarism is of course abhorrent, we object strongly to these meritless accusations being leveled against us.
>
> ### Kernel caching logic & skip connection
> Another two pieces of *evidence* provided are the techniques of caching the pre-computed kernel and the skip connection in EMA.
> First, the idea of caching a pre-computed kernel is simple and straight-forward. It is not hard (even very easy) for a graduate researcher to implement this caching technique to slightly accelerate model inference speed. Second, the `so-called` skip connection is just the residual connection (multiplied with a weight vector), which has been widely applied in various neural architectures since ResNet, including the Transformers we all are very familiar with. We’re quite surprised that this residual connection in our model could be thought to be derived from SSMs.
>
> ### S4D: Vandermonde kernel
> Albert claimed that the term *vander* in our code is somehow a *dead giveaway*, because the term *Vandermonde* only appeared in S4D.
> We used `vander` in our EMA implementation because EMA needs to compute the `Vandermonde Matrix`. `Vandermonde Matrix` is a well-known concept in linear algebra, and the Wikipedia link is here: https://en.wikipedia.org/wiki/Vandermonde_matrix
> We would disagree with the fact that usage of a well-known term in linear algebra is a *dead giveaway* that EMA is based on S4D implementation.
>
> To further prove this, we provide the clip of our git commit history (https://anonymous.4open.science/r/ICLR23_proofs-08DB/vander_commit_March_25.png), where we showed that as early as March 25, we implemented the `arange-exp-einsum` pattern to replace PyTorch’s own Vandermonde Matrix (yes, Vandermonde Matrix is so well-known that PyTorch has provided an internal implementation), because we found PyTorch’s implementation is somehow numerically unstable. We have already started to use the term `vander` in our code, which is clearly not based on the `Vandermonde` kernel in S4D. We can provide our original development repository to support that our EMA module was developed and finalized much earlier before S4D was published on arXiv but this is during the anonymity period and we couldn’t do so. We can share our original dev repo link with the full timestamped commit history of the EMA component to prove this after the anonymity period.

---

> ### Public Comment · ~Albert_Gu1 · 2023-02-23
> **Response**
>
> I sincerely appreciate the authors' response clarifying their views. The authors and I have since corresponded directly and resolved this discussion. This includes a technical improvement to the bidirectional kernel that is both fast and preserves correctness without the off-by-one. I (and other researchers) have also experimented with variations of these models and replicated the findings that simplified versions of S4, including pure EMA kernels, can be better on various settings (e.g. when combined with strong attention as in the full Mega model, or perhaps for language modalities in general).
>
> I also want to clarify that my original intention was not to accuse the authors of literally copying code - as they pointed out, none of the code examples in isolation is a big deal - but to portray that as a whole, I had reason to believe that Mega was influenced by the S4 line of work more than was depicted in the paper. This was not only executed poorly, but it's evident that my assumptions about Mega were misplaced. I'm very sorry that the authors had to resort to showing commit proof of their work.
>
> Overall, it's clear that Mega's EMA and some of the S4 followups were similar models developed concurrently, and with different interpretations and motivations. This perhaps speaks to the richness of this space and importance of their ideas. I agree with the authors that it would be great to have follow-ups to understand, consolidate, and improve these models in potential future work.

---

### Decision · Program_Chairs · 2023-01-20

**Decision:**

Accept: poster

**Justification For Why Not Higher Score:**

There are still some concerns with respect to clarity and positioning of the paper, which doesn't justify a higher score.

**Justification For Why Not Lower Score:**

The reviewers who raise such concerns seem either satisfied with the author response and discussions with other reviewers, or do not strongly advocate for rejection.

**Metareview: Summary, Strengths And Weaknesses:**

The paper proposes a new mechanism called the moving average equipped gated attention (MEGA) for addressing long-range modeling in transformers. MEGA is based on exponentially weighted moving averages (EMA) and shows improvements over various baselines on long range tasks.

All reviewers agree that the technical contribution is well motivated and the empirical results are strong, showing significant improvement across a wide range of tasks/datasets.

There has been some concerns in the reviews with respect to the novelty, clarity, theoretical results, and positioning of the the paper with respect to other closely relevant papers. However, most of the concerns have been resolved after extensive discussions with the authors.
(I would like to thank the reviewers and authors for the detailed discussions).

**Note From Pc:**

if the above contains the word "oral" or "spotlight" please see: "oral" presentation means -> notable-top-5% and "spotlight" means -> notable-top-25%. As stated in our emails, we are disassociating presentation type from AC recommendations